# LogoRA: Local-Global Representation Alignment for Robust Time Series Classification

## Abstract

Unsupervised domain adaptation (UDA) of time series aims to teach models to identify consistent patterns across various temporal scenarios, disregarding domain-specific differences, which can maintain their predictive accuracy and effectively adapt to new domains. However, existing UDA methods struggle to adequately extract and align both global and local features in time series data. To address this issue, we propose the **Lo**cal-**Gl**obal **R**epresentation **A**lignment framework (LogoRA), which employs a two-branch encoder—comprising a multi-scale convolutional branch and a patching transformer branch. The encoder enables the extraction of both local and global representations from time series. A fusion module is then introduced to integrate these representations, enhancing domain-invariant feature alignment from multi-scale perspectives. To achieve effective alignment, LogoRA employs strategies like invariant feature learning on the source domain, utilizing triplet loss for fine alignment and dynamic time warping-based feature alignment. Additionally, it reduces source-target domain gaps through adversarial training and per-class prototype alignment. Our evaluations on four time-series datasets demonstrate that LogoRA outperforms strong baselines by up to 12.52%, showcasing its superiority in time series UDA tasks.

## 1 Introduction

Time series data is found ubiquitously across various domains, including finance, healthcare, cloud computing, and environmental monitoring (Koh et al., 2021; Wen et al., 2022). Recently, deep learning techniques have demonstrated impressive capabilities in handling various time-series datasets separately (Ravuri et al., 2021). However, the challenge arises when attempting to deploy models trained on a specific source domain to tackle uncharted target domains, leading to a noticeable performance drop due to domain shifts (Purushotham et al., 2016). This issue underscores the critical importance of applying unsupervised domain adaptation (UDA) within the realm of time series. In the context of time series analysis, UDA aims to teach models to identify consistent patterns across various temporal scenarios, disregarding domain-specific differences, which ensures that the model can maintain its predictive accuracy and effectively adapt to new domains (Ozyurt et al., 2023).

Unlike supervised approaches that rely on labeled target data, UDA leverages the wealth of information contained within the source domain and exploits it to align the distributions of source and target data in the temporal domain. Prior research endeavors in this field have employed specialized feature extractors to capture the temporal dynamics inherent in multivariate time series data. These extractors commonly rely on recurrent neural networks (RNNs) (Purushotham et al., 2016), long short-term memory (LSTM) networks (Cai et al., 2021), as well as convolutional neural networks (CNNs) (Liu and Xue, 2021; Wilson et al., 2020; He et al., 2023). Other methods (Yue et al., 2022; Ozyurt et al., 2023) utilize contrastive learning to extract domain-invariant information from source domain data.

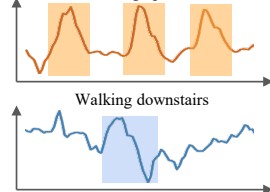

Figure 1: **A motivation example**, which contains accelerometer data pieces of walking upstairs (upper) and walking downstairs (lower) from HAR dataset.

As far as we know, none of these methods is able to adequately extract global and local features from time series data and align them across different domains. Most existing approaches (Ozyurt et al., 2023) employ a **temporal convolutional network (TCN) (Bai et al., 2018)** as the backbone and use the feature of the last time step for classification. However, some intermediate time steps may contain more valuable information.

As shown in the lower figure in Figure 1, compared to other regions, the abrupt acceleration changes in the shaded area are more representative of the class characteristics (walking downstairs) for the entire sequence, which underscores the significance of local features[1]. At the same time, focusing only on small-grained features may cause some failure cases because of ignoring long-distance feature dependencies. For instance, if only focusing on the shaded area of the upper figure in Figure 1, it is hard to distinguish the two classes (walking downstairs and walking upstairs), as both sequences have similar local features. While it is much easier to differentiate them through modeling the temporal dependencies between local features. Therefore, it is crucial to capture both global contextual features as well as local features in order to extract the discriminative features from time series.

In this paper, we propose a novel framework for unsupervised domain adaptation of time series data, called LogoRA. Considering the network architecture, we employ a two-branch encoder, using a multi-scale convolutional branch and a patching transformer branch. The convolutional network can learn local features through convolutions (Cui et al., 2016) and the transformer reflects long-distance feature dependencies by self-attention mechanism (Zhou et al., 2021). Therefore, the LogoRA is able to extract local and global representations from time series instances. We then introduce a fusion module to integrate local and global representations and make the final feature for time series classification more representative and discriminative.

With these representations, LogoRA can better align features across different domains from multi-scale perspectives. We design the following strategies to attain such a target: (1) *invariant feature learning on source domain*: we first align patch embeddings by introducing a shortest path loss based on Dynamic Time Warping (DTW) (Müller, 2007), where the alignment strategy can enable the final feature more robust to time-step shift; then we employ triplet loss for finer alignment of the fused classification features of each class; (2) *reducing source-target domain gaps*: at domain level, we minimize the domain discrepancy between source and target domains through adversarial training; at class level, we introduce a per-class center loss, which reduces the distance between target domain samples and its nearest source domain prototype. **The contributions are:**

- We propose a novel network architecture (LogoRA) comprising a multi-scale local encoder utilizing a convolutional network with different kernel sizes, a global encoder employing a transformer, and a fusion module. As far as we know, the LogoRA is the first UDA structure that learns a contextual representation considering both local and global patterns.

- We design a new metric learning method based on DTW, which can overcome the severe time-shift patterns that exist in time series data and learn more robust features from the source domain. Besides, we employ adversarial learning and introduce per-class prototype-alignment strategies to align representations between the target domain and source domain, in order to acquire domain-invariant contextual information.

- We evaluate LogoRA on four time-series datasets: HHAR, WISDM, HAR, and Sleep-EDF. Our method outperforms strong baselines by up to $12.52\%$. Besides, extensive empirical results verify the efficiency of our each design choice for the time series UDA task. Insightful visual studies are given to explain the success reasons of our algorithm.

## 2   PROBLEM DEFINITION

Unsupervised Domain Adaptation (UDA) addresses the challenge of transferring knowledge learned from a source domain $\mathcal{D}_S$ with labeled data to a target domain $\mathcal{D}_T$ where the label information is unavailable. We use $\mathcal{S} = \{(x_i^s, y_i^s)\}_{i=1}^{N_s} \sim \mathcal{D}_S$ to denote the source domain dataset with $N_s$ labeled *i.i.d.* samples, where $x_i^s$ is a source domain sample and $y_i^s$ is the associated label. Meanwhile, the target domain dataset is unlabeled and denoted as $\mathcal{T} = \{(x_i^t)\}_{i=1}^{N_t} \sim \mathcal{D}_T$.

In this paper, we consider the classification task for multivariate time series. Hence, each sample $x_i \in \mathbb{R}^{T \times d}$ (either from the source or target domain) contains $d$ observation over $T$ time steps. During the training phase, labels for target samples $\mathcal{T}$ are inaccessible. Our aim is to learn domain-invariant and contextual information from labeled source samples $\mathcal{S}$ and unlabeled target samples $\mathcal{T}$. After training, we use labeled target samples $\mathcal{T}_{\text{test}} = \{(x_i^t, y_i^t)\}_{i=1}^{N_t} \sim \mathcal{D}_T$ only for evaluation.

---

[1]Because the signal in the figure is relatively stable in other positions. If we only use the features obtained by average pooling or the features from the last time step as the classification features, it may be unable to capture this sudden signal change. This oversight could lead to a failure in recognizing this action.

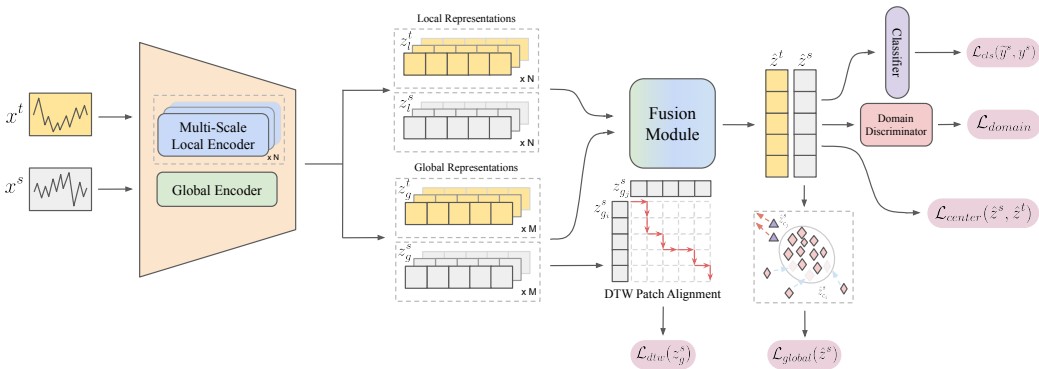

Figure 2: **Model Architecture and Training Pipeline of LogoRA.** The time series data is processed through a feature extractor, comprising a Global Encoder and a Multi-Scale Local Encoder, to extract local and global representations. Next, these representations are fed into the Fusion Module to obtain the fused representations. We further use different representations for invariant feature learning ($\mathcal{L}_{dtw}$ and $\mathcal{L}_{global}$) and alignment across the source and target domain ($\mathcal{L}_{domain}$ and $\mathcal{L}_{center}$).

## 3 LOGORA

In this section, we start with an overview of the proposed LogoRA framework and proceed with model architecture (Sec 3.2 and Sec 3.3), and feature alignment (Sec 3.4 and Sec 3.5).

### 3.1 OVERVIEW

LogoRA represents a robust unsupervised approach for mining and aligning both local and global information within time series data. The complete LogoRA framework for unsupervised domain adaptation is illustrated in Figure 2. It primarily comprises four modules: a feature extractor denoted as $F(\cdot)$, a fusion module denoted as $G(\cdot)$, a classifier represented by $C(\cdot)$, and a domain discriminator labeled as $D(\cdot)$. Sec. 3.2 elaborates on the specifics of the feature extractor, which utilizes source and target samples to yield global representations $z_g$ and local representations $z_l$ through the Global Encoder $F_g(\cdot)$ and the Multi-Scale Local encoder $F_l(\cdot)$. Then these representations are fed into the Local-Global Fusion Module to obtain the fused representations $\hat{z}$, as explained in Sec. 3.3. In the end, Sec. 3.4 details the invariant feature learning on the source domain and Sec. 3.5 details the alignment across domain representations. For symbol clarity, we conclude all the used mathematical symbols and corresponding notions in Appendix Table. 6.

### 3.2 FEATURE EXTRACTOR

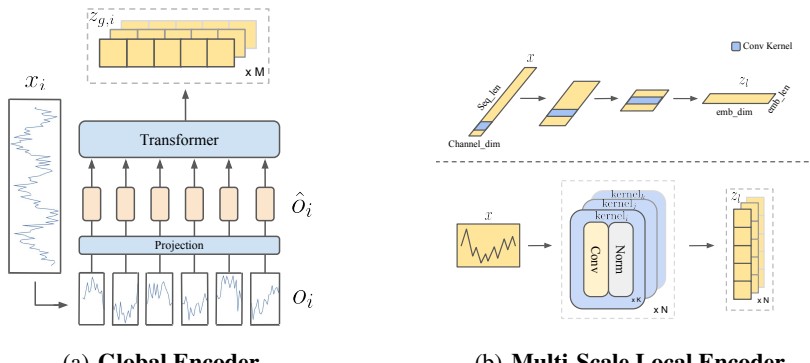

(a) **Global Encoder**

(b) **Multi-Scale Local Encoder**

Figure 3: **(a) Global Encoder:** We use a Transformer encoder with a patching operation to obtain fine-grained global representations. **(b) Multi-Scale Local Encoder:** We use ConvNet with different kernel sizes to acquire multi-scale local representations.

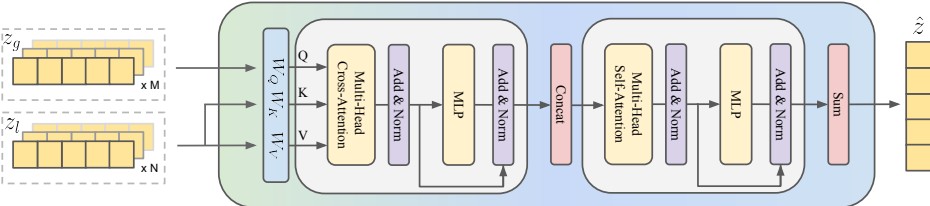

Figure 4: **Local-Global Fusion Module.** We use cross-attention to fuse global and multi-scale local representations. Next, we concatenate all the cross-attentions and sum them to get the final output.

**Global Encoder.** The architecture of Global Encoder is illustrated in Figure 3(a). Inspired by PathTST (Nie et al., 2023), each input multivariate time series $x_i \in \mathbb{R}^{T \times d}$ is first divided into several patches which can be either overlapped or non-overlapped. Let the patch length be denoted as $P$, and the non-overlapping stride between adjacent patches be denoted as $S$. Following the patching process, we obtain a sequence of patches $o_i \in \mathbb{R}^{M \times P \times d}$, where $M$ represents the number of patches for the original sequence $x_i$, and $M = \lfloor \frac{(T-P)}{S} \rfloor + 1$. Before patching, $S$ repeated numbers of the last time step value are padded to the end of the original sequence.

We use a vanilla Transformer encoder that maps the original sequence to the latent representations. The patches are mapped to the Transformer latent space of dimension $D$ via a trainable linear projection, resulting in $o'_i \in \mathbb{R}^{M \times P \times D}$. Next, self-attention is computed within each patch and then aggregated, yielding $o''_i \in \mathbb{R}^{M \times D}$. Besides, a learnable position encoding $W_{pos} \in \mathbb{R}^{M \times D}$ is also applied to monitor the temporal order of patches to get patch embeddings: $\hat{o}_i = o''_i + W_{pos}$, each of which represents a segment of the original sequence. Finally, the patch embeddings $\hat{o}_i$ are fed into Transformer to generate the global representations $z_{g,i} \in \mathbb{R}^{M \times D}$.

**Multi-Scale Local Encoder.** The Multi-Scale Local Encoder comprises $N$ convolutional neural networks (CNNs) with varying kernel sizes ($Kernel_i, i = 1, ..., N$), where each CNN consists of $K$ stages. As depicted in the lower part of Figure 3(b), each stage consists of a convolution operation followed by batch normalization. With each stage, a shorter yet deeper embedding is obtained to facilitate the preservation of more comprehensive local information, which is shown in the upper part of Figure 3(b). In summary, given an input time series $x \in \mathbb{R}^{T \times d}$, the Multi-Scale Local Encoder generates $N$ local representations $z_l = \{ z_l^{(i)} \in \mathbb{R}^{l_{\text{emb}}^{(i)} \times d_{\text{emb}}} \mid i = 1, ..., N \}$, where $l_{\text{emb}}$ and $d_{\text{emb}}$ represent the length and dimension of the output embedding from the final stage.

### 3.3 Local-Global Fusion Module

Given the local and global representations $z_l, z_g$ of the time series instances, we propose the Local-Global Fusion Module to further integrate them into unified representations, as depicted in Figure 4.

The core of the fusion module is the utilization of cross-attention mechanism to facilitate interaction between local and global representations. Similar to self-attention, the input $z_g$ and $z_l$ are first projected into three vectors respectively, *i.e.* queries $Q \in \mathbb{R}^{M \times d_k}$, keys $K \in \mathbb{R}^{l_{\text{emb}} \times d_k}$ and values $V \in \mathbb{R}^{l_{\text{emb}} \times d_v}$, where $d_k$ and $d_v$ indicate the dimensions of them. Notably, for each local representation $z_l^{(i)}$ in $z_l$, there exits a corresponding key $K^{(i)} \in \mathbb{R}^{l_{\text{emb}}^{(i)} \times d_k}$ and value $V^{(i)} \in \mathbb{R}^{l_{\text{emb}}^{(i)} \times d_v}$. In order to maximize the incorporation of global information and diverse scale local information, we compute the cross-attention between global representations $z_g$ and each local representation $z_l^{(i)}$. The specific calculation process is as follows:

$$Attn_{cross}^{(i)}(Q, K^{(i)}, V^{(i)}) = \text{Softmax}(\frac{QK^{(i)^T}}{\sqrt{d_k}})V^{(i)} \tag{1}$$

The output $Attn_{cross} = \{ Attn_{cross}^{(i)} \in \mathbb{R}^{M \times d_v} \mid i = 1, ...N \}$ holds the same length $M$ as the number of the queries. Next, we concatenate all the cross-attention $Attn_{cross}^{(i)}$ corresponding to different scales and get a feature of length $N \cdot M$ and dimensionality $d_v$. Finally, the fused contextual representation $\hat{z}$, which combines both global and local information, is obtained by computing self-attention on the concatenated feature and subsequently summing the results.

### 3.4 Invariant Feature Learning On Source Domain

Time shifts are particularly common between different sequences in time series data (Cai et al., 2021), as depicted in Figure 5. In such cases, traditional Euclidean distance may not accurately measure the similarity between two sequences. Instead, the Dynamic Time Warping (DTW) algorithm (Müller, 2007) is well-suited for calculating distances between time series with time-step shifts. Thanks to the patching operation in Sec. 3.2, we obtain a sequence of patch embeddings $z_g^s$, each of which represents a segment of the original sequence from the source domain. To enforce the learned representation to be robust to time-step shift, we explicitly align the patch representations according to the DTW distance matrix. Specifically, we propose the *DTW Alignment Loss* $\mathcal{L}_{dtw}$ on $z_g^s$ as follows:

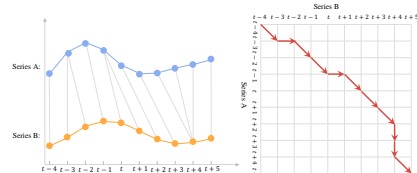

Figure 5: **Example of local distance computed by DTW.** The grey lines on the left show the corresponding alignment between two segments of time series and the red arrows on the right show the shortest path in the distance matrix.

$$\mathcal{L}_{dtw} = \sum_i^{N_s} \max(\text{DTW}(z_{g,i}^s, p) - \text{DTW}(z_{g,i}^s, n) + \alpha, 0) \tag{2}$$

where $z_{g,i}^s$ is an anchor input, $p$ is a randomly selected positive input of the same class as $z_{g,i}^s$, $n$ is a randomly selected negative input of a different class from $z_{g,i}^s$, $\alpha$ is a margin between positive and negative pairs, and $\text{DTW}(\cdot)$ represents the DTW distance. To further enhance the discrimination capability of the final fused representation, we also align global features by the proposed *Global Alignment Loss* $\mathcal{L}_{global}$ based on triplet loss. The specific definition is as follows:

$$\mathcal{L}_{global} = \sum_i^{N_s} \max(\text{dist}(\hat{z}_i^s, p) - \text{dist}(\hat{z}_i^s, n) + \beta, 0) \tag{3}$$

where $\hat{z}_i^s$ is an anchor input, $p$ is a randomly selected positive input of the same class as $\hat{z}_i^s$, $n$ is a randomly selected negative input of a different class from $\hat{z}_i^s$, $\beta$ is a margin between positive and negative pairs, and $\text{dist}(\cdot)$ represents the Euclidean distance.

### 3.5 ALIGNMENT ACROSS DOMAIN REPRESENTATIONS

Borrowing the idea from DANN (Ganin et al., 2016), we employ adversarial training for unsupervised domain adaptation by minimizing a combination of two losses. The first one is the *classification loss* $\mathcal{L}_{cls}$, which trains the feature extractor $F(\cdot)$, the Local-global Fusion Module $G(\cdot)$, and the classifier $C(\cdot)$ with the data from the source domain. Since $\mathcal{L}_{cls}$ is minimized to guarantee lower source risk, another loss $\mathcal{L}_{domain}$ is minimized over the domain discriminator $D(\cdot)$ but maximized over $F(\cdot)$, $G(\cdot)$ and $C(\cdot)$. The details of our adversarial training are described in Appendix B.

This dual optimization aims to model source and target domain features within the same feature space, with the ultimate objective of learning domain-invariant knowledge. And by enforcing the order of distances, $\mathcal{L}_{global}$ and $\mathcal{L}_{dtw}$ model embeddings with the same labels closer than those with different labels in the feature space. Consequently, we further propose the *center loss* $\mathcal{L}_{center}$ to align features of the same class between the target domain and the source domain. This is accomplished by reducing the distance between target domain samples and corresponding source domain prototypes. The definition is as follows:

$$\mathcal{L}_{center} = \sum_i^{N_t} \min_j ||\hat{z}_i^t - c_j||^2 \tag{4}$$

where $c_j$ is the $j$th class prototype in the source domain.

### 3.6 TRAINING

In summary, the overall loss of LogoRA framework is

$$\mathcal{L}_{total} = \mathcal{L}_{cls} - \lambda_{domain}\mathcal{L}_{domain} + \lambda_{global}\mathcal{L}_{global} + \lambda_{dtw}\mathcal{L}_{dtw} + \lambda_{center}\mathcal{L}_{center} \tag{5}$$

where hyper-parameters $\lambda_{domain}, \lambda_{global}, \lambda_{dtw}$ and $\lambda_{center}$ control the contribution of each component. And the complete training follows:

$$\min_{C,F,G} \mathcal{L}_{total}; \quad \min_D \mathcal{L}_{domain} \tag{6}$$

The detailed training algorithm is depicted in Appendix Alg. 1.

Table 1: **UDA performance on benchmark datasets.** Prediction accuracy for each dataset between various subjects. LogoRA consistently outperforms all other methods in accuracy on test sets drawn from the target domain dataset.

| Source ↦ Target | VRADA | CoDATS | AdvSKM | CDAN | CORAL | DSAN | HoMM | MMDA | CLUDA | RAINCOAT | **LogoRA** | Improvement |
|---|---|---|---|---|---|---|---|---|---|---|---|---|
| HHAR 0 ↦ 2 | 0.593 | 0.65 | 0.681 | 0.676 | 0.618 | 0.292 | 0.680 | 0.671 | 0.726 | 0.788 | **0.840** | +6.60% |
| HHAR 1 ↦ 6 | 0.690 | 0.686 | 0.652 | 0.717 | 0.712 | 0.689 | 0.725 | 0.686 | 0.855 | 0.889 | **0.916** | +3.04% |
| HHAR 2 ↦ 4 | 0.476 | 0.381 | 0.291 | 0.472 | 0.332 | 0.229 | 0.332 | 0.238 | 0.585 | 0.538 | **0.936** | +60.00% |
| HHAR 4 ↦ 0 | 0.263 | 0.229 | 0.203 | 0.262 | 0.259 | 0.193 | 0.205 | 0.205 | 0.353 | 0.268 | **0.389** | +10.20% |
| HHAR 4 ↦ 1 | 0.558 | 0.501 | 0.494 | 0.690 | 0.482 | 0.504 | 0.628 | 0.551 | 0.774 | 0.898 | **0.963** | +7.24% |
| HHAR 5 ↦ 1 | 0.775 | 0.761 | 0.737 | 0.857 | 0.787 | 0.407 | 0.787 | 0.790 | 0.948 | 0.977 | **0.985** | +0.82% |
| HHAR 7 ↦ 1 | 0.575 | 0.551 | 0.426 | 0.413 | 0.511 | 0.366 | 0.496 | 0.415 | 0.875 | 0.887 | **0.948** | +6.88% |
| HHAR 7 ↦ 5 | 0.523 | 0.380 | 0.192 | 0.492 | 0.489 | 0.233 | 0.328 | 0.320 | 0.636 | **0.852** | 0.815 | -0.34% |
| HHAR 8 ↦ 3 | 0.813 | 0.766 | 0.748 | 0.942 | 0.869 | 0.602 | 0.844 | 0.934 | 0.942 | 0.973 | **0.974** | +0.10% |
| HHAR 8 ↦ 4 | 0.720 | 0.601 | 0.650 | 0.712 | 0.618 | 0.516 | 0.658 | 0.701 | 0.896 | 0.796 | **0.968** | +8.04% |
| HHAR Avg | 0.599 | 0.551 | 0.508 | 0.623 | 0.568 | 0.403 | 0.567 | 0.551 | 0.759 | 0.775 | **0.872** | +12.52% |
| WISDM 12 ↦ 19 | 0.558 | 0.633 | 0.639 | 0.488 | 0.433 | 0.639 | 0.415 | 0.358 | 0.694 | 0.530 | **0.742** | +6.92% |
| WISDM 12 ↦ 7 | 0.708 | 0.721 | 0.742 | 0.771 | 0.592 | 0.625 | 0.546 | 0.679 | 0.792 | 0.875 | **0.896** | +2.40% |
| WISDM 18 ↦ 20 | 0.571 | 0.634 | 0.390 | 0.771 | 0.380 | 0.366 | 0.429 | 0.380 | 0.780 | 0.756 | **0.829** | +6.28% |
| WISDM 19 ↦ 2 | 0.644 | 0.395 | 0.434 | 0.346 | 0.473 | 0.366 | 0.488 | 0.385 | 0.561 | 0.659 | **0.756** | +14.72% |
| WISDM 2 ↦ 28 | 0.729 | 0.809 | 0.809 | 0.813 | 0.827 | 0.773 | 0.787 | 0.813 | 0.849 | 0.798 | **0.889** | +4.71% |
| WISDM 26 ↦ 2 | 0.683 | 0.727 | 0.620 | 0.615 | 0.737 | 0.605 | 0.702 | 0.634 | 0.863 | 0.598 | **0.878** | +1.74% |
| WISDM 28 ↦ 2 | 0.688 | 0.717 | 0.707 | 0.580 | 0.649 | 0.673 | 0.644 | 0.668 | 0.741 | 0.585 | **0.854** | +15.25% |
| WISDM 28 ↦ 20 | 0.741 | 0.741 | 0.707 | 0.776 | 0.737 | 0.746 | 0.790 | 0.722 | 0.820 | 0.804 | **0.927** | +13.05% |
| WISDM 7 ↦ 2 | 0.605 | 0.610 | 0.610 | 0.649 | 0.624 | 0.620 | 0.605 | 0.605 | 0.712 | **0.817** | 0.781 | -4.41% |
| WISDM 7 ↦ 26 | 0.693 | 0.702 | 0.702 | 0.722 | 0.683 | 0.698 | 0.698 | 0.712 | 0.727 | 0.732 | **0.756** | +3.28% |
| WISDM Avg | 0.662 | 0.669 | 0.636 | 0.653 | 0.713 | 0.611 | 0.610 | 0.596 | 0.754 | 0.715 | **0.831** | +10.21% |
| Sleep-EDF 0 ↦ 11 | 0.499 | 0.695 | 0.565 | 0.689 | 0.572 | 0.518 | 0.278 | 0.245 | 0.579 | 0.744 | **0.746** | +0.27% |
| Sleep-EDF 2 ↦ 5 | 0.578 | 0.718 | 0.656 | 0.695 | 0.604 | 0.422 | 0.455 | 0.438 | 0.719 | 0.738 | **0.754** | +2.17% |
| Sleep-EDF 12 ↦ 5 | 0.655 | 0.793 | 0.765 | 0.785 | 0.750 | 0.434 | 0.399 | 0.471 | 0.794 | 0.798 | **0.849** | +6.39% |
| Sleep-EDF 7 ↦ 18 | 0.671 | 0.732 | 0.609 | 0.732 | 0.658 | 0.360 | 0.549 | 0.533 | 0.745 | 0.753 | **0.770** | +2.26% |
| Sleep-EDF 16 ↦ 1 | 0.798 | 0.753 | 0.730 | 0.745 | 0.695 | 0.534 | 0.507 | 0.547 | 0.758 | **0.786** | 0.748 | -7.99% |
| Sleep-EDF 9 ↦ 14 | 0.733 | 0.816 | 0.768 | 0.801 | 0.822 | 0.503 | 0.469 | 0.504 | 0.863 | **0.872** | 0.862 | -1.15% |
| Sleep-EDF 4 ↦ 12 | 0.576 | **0.717** | 0.661 | 0.671 | 0.415 | 0.424 | 0.435 | 0.668 | 0.665 | 0.699 | 0.688 | -4.04% |
| Sleep-EDF 10 ↦ 7 | 0.570 | 0.733 | 0.743 | 0.734 | 0.761 | 0.529 | 0.517 | 0.526 | 0.752 | 0.772 | **0.797** | +3.24% |
| Sleep-EDF 6 ↦ 3 | 0.751 | 0.836 | 0.789 | 0.810 | 0.784 | 0.531 | 0.510 | 0.506 | 0.820 | **0.846** | 0.841 | -0.59% |
| Sleep-EDF 8 ↦ 10 | 0.458 | 0.442 | 0.448 | 0.552 | 0.368 | 0.544 | 0.501 | 0.436 | 0.657 | 0.624 | **0.754** | +5.90% |
| Sleep-EDF Avg | 0.629 | 0.724 | 0.673 | 0.722 | 0.667 | 0.461 | 0.479 | 0.464 | 0.735 | 0.763 | **0.781** | +2.36% |
| HAR 15 ↦ 19 | 0.756 | 0.733 | 0.741 | 0.759 | 0.759 | 0.874 | 0.748 | 0.726 | 0.967 | 1.000 | **1.000** | +0.00% |
| HAR 18 ↦ 21 | 0.794 | 0.522 | 0.555 | 0.803 | 0.610 | 0.558 | 0.581 | 0.555 | 0.910 | 1.000 | **1.000** | +0.00% |
| HAR 19 ↦ 25 | 0.768 | 0.468 | 0.452 | 0.771 | 0.590 | 0.774 | 0.487 | 0.448 | **0.932** | 0.885 | 0.887 | -4.83% |
| HAR 19 ↦ 27 | 0.793 | 0.709 | 0.723 | 0.807 | 0.744 | 0.891 | 0.726 | 0.754 | 0.996 | 0.989 | **1.000** | +0.40% |
| HAR 20 ↦ 6 | 0.808 | 0.661 | 0.641 | 0.820 | 0.686 | 0.784 | 0.673 | 0.694 | 1.000 | 0.989 | **1.000** | +0.00% |
| HAR 23 ↦ 13 | 0.736 | 0.504 | 0.504 | 0.700 | 0.688 | 0.628 | 0.604 | 0.572 | 0.778 | 0.885 | **0.920** | +3.62% |
| HAR 24 ↦ 22 | 0.837 | 0.820 | 0.833 | 0.837 | 0.743 | 0.808 | 0.853 | 0.829 | 0.988 | 1.000 | **1.000** | +0.00% |
| HAR 25 ↦ 24 | 0.817 | 0.583 | 0.566 | 0.790 | 0.648 | 0.883 | 0.607 | 0.666 | 0.993 | 1.000 | **1.000** | +0.00% |
| HAR 3 ↦ 20 | 0.752 | 0.874 | 0.878 | 0.815 | 0.848 | 0.804 | 0.874 | 0.815 | 0.967 | **1.000** | 0.982 | -0.18% |
| HAR 13 ↦ 19 | 0.752 | 0.793 | 0.807 | 0.841 | 0.793 | 0.726 | 0.815 | 0.800 | 0.967 | 1.000 | **1.000** | +0.00% |
| HAR Avg | 0.781 | 0.670 | 0.670 | 0.794 | 0.709 | 0.773 | 0.697 | 0.686 | 0.944 | 0.974 | **0.979** | +0.51% |

## 4 EXPERIMENTS

### 4.1 EXPERIMENTAL SETUP

**Dataset:** We consider four benchmark datasets from multiple modalities: WISDM (Kwapisz et al., 2011), HAR (Anguita et al., 2013), HHAR (Stisen et al., 2015), Sleep-EDF (Goldberger et al., 2000). Further details on datasets are given in Appendix C. Following previous work on DA for time series (Ozyurt et al., 2023; He et al., 2023), we select the same ten pairs of domains to specify source ↦ target domains.

**Baseline:** We compare the performance of our LogoRA on unsupervised domain adaptation with five state-of-the-art baselines for UDA of time series: VRADA (Purushotham et al., 2016), Co-DATS (Wilson et al., 2020), AdvSKM (Liu and Xue, 2021), CLUDA (Ozyurt et al., 2023) and RAINCOAT (He et al., 2023). We also consider five general UDA methods for a comprehensive comparison: CDAN (Long et al., 2018), DeepCORAL (Sun and Saenko, 2016), DSAN (Zhu et al., 2020), HoMM (Chen et al., 2020) and MMDA (Rahman et al., 2020). The implementation details are described in Appendix Section D.

**Evaluation:** We report the mean accuracy calculated on target test datasets. The accuracy is computed by dividing the number of correctly classified samples by the total number of samples.

### 4.2 NUMERICAL RESULTS ON UDA BENCHMARKS

Following previous work, we present the prediction results for 10 source-target domain pairs for each dataset in Table 1. Overall, LogoRA has won 4 out of 4 datasets and makes an average improvement of accuracy (6.40%) over with the strongest baseline across datasets. Specifically, on HHAR dataset, our LogoRA outperforms the best baseline accuracy of RAINCOAT by 12.52% (0.872 vs.

0.775). On WISDM dataset, our LogoRA outperforms the best baseline accuracy of CLUDA by 10.21% (0.831 vs. 0.754). On Sleep-EDF dataset, our LogoRA outperforms the best baseline accuracy of RAINCOAT by 2.36% (0.781 vs. 0.763). And on HAR dataset, our LogoRA outperforms the best baseline accuracy of RAINCOAT by 0.51% (0.979 vs. 0.974). The overall performance demonstrates that LogoRA successfully extracts both global and local information from sequences, learning domain-invariant features and achieving alignment across different domains. This enhances knowledge transfer for time series data in the presence of domain shifts.

## 4.3  ABLATION STUDIES

Table 2: **Ablation studies of loss function.** Specifically, the loss functions, $\mathcal{L}_{cls}$, $\mathcal{L}_{domain}$, $\mathcal{L}_{global}$, $\mathcal{L}_{dtw}$, and $\mathcal{L}_{center}$, are shown below. When only the classification loss $\mathcal{L}_{cls}$ is used (first row), it refers to a source-only model, which is trained exclusively on the source domain. We evaluate LogoRA across 10 scenarios on the HHAR dataset and report the mean Accuracy.

| Element of LogoRA | | | | | UDA performance | | | | |
|---|---|---|---|---|---|---|---|---|---|
| $\mathcal{L}_{cls}$ | $\mathcal{L}_{domain}$ | $\mathcal{L}_{global}$ | $\mathcal{L}_{dtw}$ | $\mathcal{L}_{center}$ | $2 \mapsto 4$ | $4 \mapsto 1$ | $7 \mapsto 1$ | $8 \mapsto 3$ | Avg (10 scenarios) |
| ✓ | | | | | 0.514 | 0.772 | 0.743 | 0.891 | 0.707 |
| ✓ | ✓ | | | | 0.602 | 0.825 | 0.888 | 0.974 | 0.796 |
| ✓ | | ✓ | ✓ | | 0.490 | 0.784 | 0.791 | 0.843 | 0.717 |
| ✓ | ✓ | | | ✓ | 0.438 | 0.948 | 0.914 | 0.974 | 0.792 |
| ✓ | ✓ | ✓ | | ✓ | 0.892 | 0.806 | 0.696 | 0.974 | 0.787 |
| ✓ | ✓ | | ✓ | ✓ | 0.546 | 0.847 | 0.862 | 0.974 | 0.764 |
| ✓ | ✓ | ✓ | ✓ | ✓ | **0.896** | **0.970** | **0.933** | **0.974** | **0.872** |

**Ablation studies of loss function:** To verify the effectiveness of invariant feature learning and alignment across diverse domains, we conduct an ablation study on the challenging HHAR dataset, presenting the results in Table 2. In the first row, where only the classification loss $\mathcal{L}_{cls}$ is used, we refer to it as a 'source-only' model, trained exclusively on the source domain. In the second row, when employing only adversarial learning strategies, a noticeable improvement in performance on the target domain is observed. This indicates that adversarial learning enables the model to initially learn some domain-invariant features, aligning the target domain with the source domain. When the center loss $\mathcal{L}_{center}$ is further applied (4th row in Table 2), there are significant improvements in specific source-target domain pairs (e.g., $4 \mapsto 1$, $7 \mapsto 1$, $8 \mapsto 3$), yet a decrease in performance is noted in others (e.g., $2 \mapsto 4$). Consequently, the final average accuracy remains relatively stable. This suggests that features from the source domain are not adequately aligned in the feature space, affecting the center loss's effectiveness. When aligning them using the global loss, the UDA accuracy for the source-target domain pair $2 \mapsto 4$ sees a significant improvement (5th row in Table 2). Besides, learning invariant features and aligning global features only in the source domain (3rd row in Table 2) leads to a slight improvement in performance. When additionally aligning target domain features with source domain features (7th row in Table 2), *i.e.* our LogoRA, the performance of UDA is improved by a large margin. To study the impact of the global loss $\mathcal{L}_{global}$ and the DTW loss $\mathcal{L}_{dtw}$, we separately test the prediction results by removing each one (5th and 6th row in Table 2). The results show a varying degree of decrease in accuracy compared to the original model, which indicates that both global and DTW loss are indispensable, and they play crucial roles in the entire framework. Without the global loss, the features learned by the model may not be sufficiently aligned in the feature space, thereby impacting the effectiveness of the center loss. Namely, without the DTW loss, the model fails to adequately learn features robust to time-step shifts, resulting in decreased accuracy.

**Ablation studies of model architecture:** In order to explore different model architectures and investigate their generalization capabilities, we conduct a series of ablation experiments on the source domain. All models are trained only on source domain data using the classification loss $\mathcal{L}_{cls}$, and ultimately tested on the target domain to isolate their performance from other variables. We perform the ablation study using HHAR dataset and present results in Table 3. We initially attempt the commonly used backbone, TCN (Bai et al., 2018), in time series analysis (1st row in Table 3). It was evident that TCN exhibits significantly lower generalization capability compared to other backbones. We believe this might be attributed to TCN's dilated convolution mechanism, which may not effectively extract information from continuous time series data. Next, we attempt using vanilla Transformer and PatchTST (Nie et al., 2023) as backbones (2nd and 3st row in Table 3). Both show

Table 3: **Ablation studies of model architecture.** We evaluate different backbones across 10 scenarios on the HHAR dataset and report the mean Accuracy. All the models are only trained to minimize the classification loss on the source domain.

| Model | UDA performance | | | | |
|---|---|---|---|---|---|
| | $2 \mapsto 4$ | $4 \mapsto 1$ | $7 \mapsto 1$ | $8 \mapsto 3$ | Avg (10 scenarios) |
| TCN | 0.296 | 0.454 | 0.358 | 0.358 | 0.496 |
| Transformer | 0.502 | 0.563 | 0.567 | 0.852 | 0.601 |
| PatchTST | 0.442 | 0.683 | 0.731 | 0.834 | 0.629 |
| Transformer + TCN | 0.463 | 0.689 | 0.610 | 0.659 | 0.619 |
| Global Encoder + TCN | 0.346 | 0.560 | 0.739 | 0.865 | 0.623 |
| Global Encoder + Local Encoder | 0.406 | 0.716 | 0.582 | 0.825 | 0.633 |
| LogoRA | **0.514** | **0.772** | **0.743** | **0.891** | **0.707** |

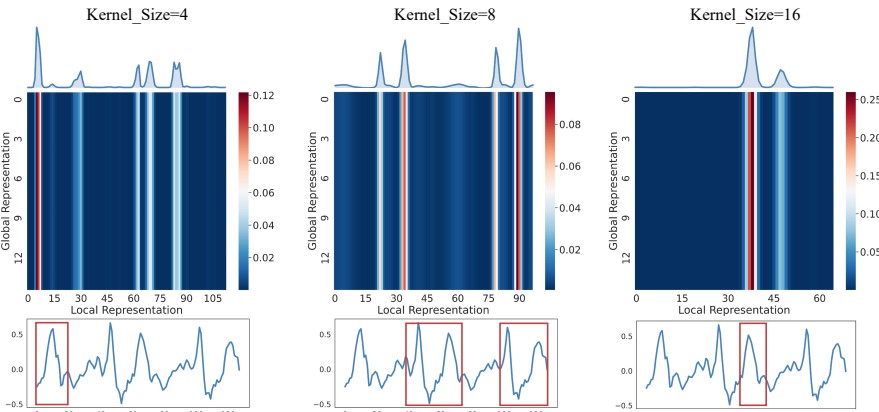

Figure 6: **Cross-Attention Heat Map:** The heatmap displays the cross-attention weights calculated between global representation and multi-scale local representation within the fusion module. The horizontal and vertical axes represent the lengths of the local representation and global representation, respectively. From left to right, the results are shown for kernel sizes of 4, 8, and 16, respectively, on the HAR dataset. The figure below represents the specific data from one channel of the corresponding original time series. Clearly, thanks to the multiscale operation, LogoRA is able to thoroughly attend to sufficiently detailed local features.

significant improvements compared to TCN, indicating the superior effectiveness of the Transformer architecture. Notably, PatchTST outperforms the standard Transformer, highlighting the advantage of patching to direct the attention of the Transformer toward local information. Subsequently, we experiment with the two-branch architecture (4th to 7th row in Table 3), where each branch specializes in extracting either global or local features. When employing our global encoder and local encoder (6th row in Table 3), the performance surpasses that of the previous models. Furthermore, with the adoption of the multi-scale local encoder, which corresponds to our LogoRA, the performance is the best, demonstrating a noticeable enhancement in generalization.

### 4.4 VISUALIZATION

**Visualizations of cross-attention from fusion module:** For a more intuitive understanding of the impact of multi-scale operations and to confirm whether LogoRA has extracted meaningful features from the time series data, we visualize the cross-attention weights computed between local representation and global representation within the fusion module. Taking the HAR dataset as an example, in Figure 6, the middle section displays the heat map of attention weights. The horizontal and vertical axes represent the lengths of local representation and global representation, respectively. Above, the waveform chart illustrates the average weight at each position of the local representation. From left to right, the results are shown for kernel sizes of 4, 8, and 16, representing different scale local encoder respectively. The figure below represents the specific data from one channel of the corresponding original time series. The locations with higher attention weights are highlighted in the original data. Clearly, thanks to the multi-scale operation, LogoRA is able to thoroughly attend

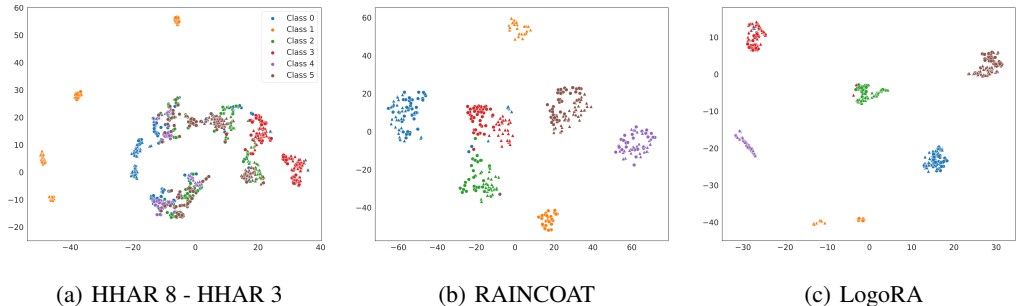

(a) HHAR 8 - HHAR 3        (b) RAINCOAT        (c) LogoRA

Figure 7: **T-SNE plots of different methods and raw data on the HHAR dataset of adapting from source 8 to target 3**. Each color corresponds to a different class. The circle markers represent source samples, while the triangle markers represent target samples. These T-SNE plots demonstrate that LogoRA concentrates features from the same category more effectively and aligns the target domain with the source domain more accurately.

to sufficiently detailed local features, which not only aids in learning meaningful features but also facilitates the alignment between features.

**T-SNE visualizations of learned representations:** We generate T-SNE plots of learned embeddings for different methods. In Figure 7, we present the T-SNE plots of the original data, RAIN-COAT (He et al., 2023) and our LogoRA for HHAR dataset of adapting source 8 to target 3. Despite the original data being challenging to discern (Figure7(a)), LogoRA is still capable of effectively distinguishing between different categories and aligning the features of the target domain with the source domain. Additionally, the T-SNE plots reveal that the clusters in Figure 7(c) are generally more tightly grouped and better separated compared to those in Figure 7(b), which demonstrates the effectiveness of our proposed loss functions. Specifically, for Class 1, which experiences significant domain shift, LogoRA achieves a much better alignment than other methods. This suggests that LogoRA effectively adapts the model to the target domain, leading to improved performance and more accurate predictions. These findings also demonstrate the efficacy of LogoRA for domain adaptation and highlight its potential for a wide range of applications, including robotics, healthcare, and sports performance analysis. T-SNE plots of other methods are shown in Figure 9 in Appendix Section F.3.

We discuss the inference time and model parameters in Appendix Section F.1, and analyze failure cases in Appendix Section F.4. We also conduct detailed ablation studies of each important hyperparameter in Appendix Section F.2.

## 5 CONCLUSION AND FUTURE WORK

Through an investigation of previous works on UDA for time series, we find that existing methods do not sufficiently explore global and local features as well as the alignments in time series, thereby limiting the model's generalization capability. To this end, we propose LogoRA, a novel framework that takes advantage of both global and local representations of time series data. Furthermore, we devise loss functions based on DTW and triplet loss to learn time-shift invariant features in the source domain. We then employ adversarial training and metric-based methods to further align features across different domains. As a result, LogoRA achieves the state-of-the-art performance. Extensive ablation studies demonstrate the fusion of global and local representations and the alignment losses both yield clear performance improvements. The visualization results intuitively demonstrate the impact of multi-scale operations and the superior generalization capability of LogoRA.

We acknowledge that the introduction of new model architectures leads to an increase in the number of parameters (Figure 8). We believe that there is potential for new, lighter-weight models to be developed for extracting global and local features from time series. Additionally, while we have designed some alignment loss functions, we still meet some failures in the challenging practical datasets. Therefore, there is room for the development of more efficient adaptation methods.

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

# LogoRA: Local-Global Representation Alignment for Robust Time Series Classification

# —————Appendix—————

## CONTENTS

## A  RELATED WORK

**Unsupervised Domain Adaptation:** Unsupervised Domain Adaptation (UDA) has witnessed substantial progress in recent years, which leverages labeled source domain to predict the labels of an unlabeled target domain. UDA methods attempt to minimize the domain discrepancy in order to lower the bound of the target error (Ben-David et al., 2010). We organize these methods into three categories: (1) *Adversarial-based methods:* Adversarial training approaches aim to reduce domain shift by introducing a domain discriminator that encourages the model to learn domain-invariant features. The adversarial objective is to train a feature extractor that cannot be used to distinguish between source and target domains. Examples are DANN (Ganin et al., 2016), CDAN (Long et al., 2018), ADDA (Tzeng et al., 2017), DM-ADA (Xu et al., 2020), and MADA (Pei et al., 2018). (2) *Statistical divergence-based methods:* Statistical divergence-based methods focus on minimizing the distributional gap between the source and target domains. Maximum Mean Discrepancy (MMD) (Rozantsev et al., 2018) is a widely used metric for this purpose. Other examples are CORAL (Sun and Saenko, 2016), DSAN (Zhu et al., 2020), HoMM (Chen et al., 2020), and MMDA (Rahman et al., 2020). (3) *Self-supvervised-based methods:* Self-supervised learning has emerged as a promising avenue for UDA, bypassing the need for labeled target data. Approaches in this category design pretext tasks that generate surrogate labels for the target domain. Examples are CAN (Kang et al., 2019), CLDA (Singh, 2021), HCL (Huang et al., 2021)), and GRCL (Tang et al., 2021). While these computer vision-based UDA methods can be applied to time series data, they may not sufficiently extract features from time series data. In contrast, LogoRA is specifically designed for time series, simultaneously extracting both global and local features.

**Unsupervised Domain Adaptation for Time Series:** Despite the significant achievements of Domain Adaptation in other fields, there are only a few methods specifically tailored for time series data. (1) *Adversarial-based methods:* CoDATS (Wilson et al., 2020) builds upon the same adversarial training as VRADA, but uses a convolutional neural network for the feature extractor. (2) *Statistical divergence-based methods:* SASA (Cai et al., 2021) accomplishes the alignment between the associative structure of time series variables from different domains by minimizing the maximum mean discrepancy (MMD). AdvSKM (Liu and Xue, 2021) introduces a spectral kernel mapping to minimize MMD between the source and target domain. RAINCOAT (He et al., 2023) aligns both temporal and frequency features extracted from its proposed time-frequency encoder by minimizing a domain alignment loss based on Sinkhorn divergence (Cuturi and Peyré, 2016). (3) *Self-supvervised-based methods:* DAF (Jin et al., 2022) uses a shared attention module to extract domain-invariant and domain-specific features and then perform forecasts for source and target domains. CLUDA (Ozyurt et al., 2023) applies augmentations to extract domain-invariant features by contrastive learning. However, these methods have not fully explored the local and global features inherent in time series data. In contrast, LogoRA is capable of profoundly extracting and finely aligning both local and global features.

## B  ADVERSARIAL TRAINING

Borrowing the idea from DANN (Ganin et al., 2016), the classification loss $\mathcal{L}_{cls}$ trains the feature extractor $F(\cdot)$, the Local-Global Fusion Module $G(\cdot)$, and the classifier $C(\cdot)$ on the source domain, and its specific definition is as follows:

$$\mathcal{L}_{cls} = \frac{1}{N_s} \sum_i^{N_s} \text{CrossEntropy}(C(G(F(x_i^s))), y_i^s) \tag{7}$$

The loss $\mathcal{L}_{domain}$ is minimized over the domain discriminator $D(\cdot)$ but maximized over $F(\cdot), G(\cdot)$, and $C(\cdot)$:

$$\mathcal{L}_{domain} = \frac{1}{N_s} \sum_i^{N_s} \log[D(G(F(x_i^s)))] - \frac{1}{N_t} \sum_i^{N_t} \log[1 - D(G(F(x_i^t)))] \tag{8}$$

Then the complete minmax game of adversarial training in LogoRA is:

$$\min_{C,F,G} \mathcal{L}_{cls} - \lambda_{domain}\mathcal{L}_{domain}; \quad \min_D \mathcal{L}_{domain} \tag{9}$$

## C   DATASET DETAIL

We evaluate LogoRA on 4 benchmark datasets, the details of them are as follows:

**(1) HHAR** (Stisen et al., 2015): The dataset comprises 3-axis accelerometer measurements from 30 participants. These measurements were recorded at 50 Hz, and we utilize non-overlapping segments of 128 time steps to predict the participant's activity type. The activities fall into six categories: biking, sitting, standing, walking, walking upstairs, and walking downstairs.

**(2) WISDM** (Kwapisz et al., 2011): The dataset comprises 3-axis accelerometer measurements collected from 30 participants at a frequency of 20 Hz. To predict the activity label of each participant during specific time segments, we employ non-overlapping segments consisting of 128 time steps. The dataset encompasses six distinct activity labels: walking, jogging, sitting, standing, walking upstairs, and walking downstairs.

**(3) HAR** (Anguita et al., 2013): The dataset encompasses measurements from a 3-axis accelerometer, 3-axis gyroscope, and 3-axis body acceleration. This data is gathered from 30 participants at a sampling rate of 50 Hz. Like the WISDM dataset, we employ non-overlapping segments of 128 time steps for classification. The objective is to classify the time series into six activities: walking, walking upstairs, walking downstairs, sitting, standing, and lying down.

**(4) Sleep-EDF** (Goldberger et al., 2000): The dataset comprises electroencephalography (EEG) readings from 20 healthy individuals. The goal is to classify the EEG readings into five sleep stages: wake (W), non-rapid eye movement stages (N1, N2, N3), and rapid eye movement (REM). Consistent with previous research, our analysis primarily focuses on the Fpz-Cz channel.

Table 4: **Details of benchmark datasets.**

| Dataset | Subjects | Channels | Length | Class | Train | Test |
|---------|----------|----------|--------|-------|-------|------|
| HHAR | 9 | 3 | 128 | 6 | 12,716 | 5,218 |
| WISDM | 30 | 3 | 128 | 6 | 1,350 | 720 |
| HHAR | 30 | 9 | 128 | 6 | 2,300 | 990 |
| Sleep-EDF | 20 | 1 | 3000 | 2 | 160,719 | 107,400 |

## D   IMPLEMENTATION DETAILS

In this section, we provide implementation details of LogoRA and the baseline methods. The implementation was done in PyTorch. To ensure fair comparisons, we carefully selected the appropriate encoder and scale across all methods. Except for RAINCOAT (He et al., 2023), all other baselines are configured according to the experimental settings within CLUDA (Ozyurt et al., 2023). During model training, we employed the Adam optimizer for all methods, with carefully tuned learning rates specific to each method. The hyperparameters of Adam were selected after conducting a grid search on source validation datasets, exploring a range of learning rates from $1 \times 10^{-4}$ to $1 \times 10^{-2}$. The learning rates were chosen to optimize the performance of each method. For LogoRA, we use an 8-layer transformer as the global encoder and three convolutional networks with varying kernel sizes as the local encoder. In the patching operation, for patches of different lengths, we uniformly use half of their length as the stride. And the stride in local encoder is always set to be 1. The other hyperparameters for LogoRA and baselines are reported in Table 5.

Table 5: **Hyperparameter tuning.**

| Method | Hyperparameter | Tuning Range |
|---|---|---|
| All methods | Classifier hidden dim. | 32, 64, 128 |
| | Dropout | 0, 0.1, 0.2, 0.3 |
| | Weight decay | $1 \times 10^{-4}, 1 \times 10^{-3}$ |
| VRADA (Purushotham et al., 2016) | VRNN hidden dim. | 32, 64, 28 |
| | VRNN latent dim. | 32, 64, 128 |
| | VRNN num. layers | 1, 2, 3 |
| | Discriminator hidden dim. | 64, 128, 256 |
| | Weight discriminator loss | 0.1, 0.5, 1 |
| | Weight KL divergence | 0.1, 0.5, 1 |
| | Weight neg. log-likelihood | 0.1, 0.5, 1 |
| CoDATS (Wilson et al., 2020) | Discriminator hidden dim. | 64, 128, 256 |
| | Weight discriminator loss | 0.1, 0.5, 1 |
| AdvSKM (Liu and Xue, 2021) | Spectral kernel hidden dim. | 32, 64, 128 |
| | Spectral kernel output dim. | 32, 64, 128 |
| | Spectral kernel type | Linear, Gaussian |
| | Num. kernel (if Gaussian) | 3, 5, 7 |
| | Weight MMD loss | 0.1, 0.5, 1 |
| CDAN (Long et al., 2018) | Discriminator hidden dim. | 64, 128, 256 |
| | Multiplier discriminator update | 0.1, 1, 10 |
| CORAL (Sun and Saenko, 2016) | Weight CORAL loss | 0.1, 0.3, 0.5, 1 |
| DSAN (Zhu et al., 2020) | Kernel multiplier | 1, 2, 3 |
| | Num.kernel | 3, 5, 7 |
| | Weight domain loss | 0.1, 0.5, 1 |
| HoMM (Chen et al., 2020) | Moment order | 1, 2, 3 |
| | Weight domain discrepancy loss | 0.1, 0.5, 1 |
| | Weight discriminative clustering loss | 0.1, 0.5, 1 |
| MMDA (Rahman et al., 2020) | Kernel type | Linear,Gaussian |
| | Num. kernel (if Gaussian) | 3, 5, 7 |
| | Weight MMD loss | 0.1, 0.5, 1 |
| | Weight CORAL loss | 0.1, 0.5, 1 |
| | Weight Entropy loss | 0.1, 0.5, 1 |
| CLUDA (Ozyurt et al., 2023) | Momentum | 0.9, 0.95, 0.99 |
| | Queue size | 24576, 49152, 98304 |
| | Discriminator hidden dim. | 64, 128, 256 |
| | Projector hidden dim. | 64, 128, 256 |
| | $\lambda_{\text{disc}}$ | 0.1, 0.5, 1 |
| | $\lambda_{\text{CL}}$ | 0.05, 0.1, 0.2 |
| | $\lambda_{\text{NNCL}}$ | 0.05, 0.1, 0.2 |
| RAINCOAT (He et al., 2023) | Fourier frequency mode | 64, 200 |
| LogoRA (ours) | Kernel size | 4, 8, 16, 32, 64, 128 |
| | Transformer dim. | 8, 16, 32, 64, 128 |
| | patch length | 8, 16, 32, 64, 128 |
| | Transformer Num. layers | 4, 6, 8 |
| | $\lambda_{domain}$ | 1, 1.5, 2, 2.5 |
| | $\lambda_{global}$ | 0.05, 0.1, 0.3, 0.5, 0.7 |
| | $\lambda_{dtw}$ | 0.05, 0.1, 0.3, 0.5, 0.7 |
| | $\lambda_{center}$ | 0.05, 0.1, 0.3, 0.5 |

---

**Algorithm 1** Training and inference algorithm of LogoRA

---

1: **Input:** Multivariate time series $x \in \mathbb{R}^{T \times d}$ with $d$ variables and length $T$, where $x^s$ represents the data from source domain and $x^t$ represents the data from target domain. Only the label of source domain $y^t$ is accessible. The entire framework is trained for $E$ epochs.
2: **Initialize:** a global encoder $f_g$, a multi-scale local encoder $f_l$, a fusion module $g$, a classifier $f_{cls}$ and a domain discriminator $f_{domain}$.
3: **for** each $i \in [1, E]$ **do**
4:     **Get representations from two encoders.**
5:     $z_g \in \mathbb{R}^{M \times D} = f_g(x)$                   *// Global representations from global encoder*
6:     $z_l = \{z_l^{(i)} \in \mathbb{R}^{l_{emb}^{(i)} \times d_{emb}} \mid i = 1, ..., N\} = f_l(x)$   *// Local representations from local encoder*
7:     **Get fused representations from fusion module.**
8:     $\hat{z} \in \mathbb{R}^D = g(x)$                      *// Fused representations from fusion module*
9:     **Get prediction results from classifier.**
10:    $\widetilde{y^s} \in \mathbb{R}^C = f_{cls}(\hat{z}^s)$               *// Prediction results on source domain*
11:    $\widetilde{y^t} \in \mathbb{R}^C = f_{cls}(\hat{z}^t)$               *// Prediction results on target domain*
12:    **Adversarial training.**
13:    Compute: $\mathcal{L}_{domain}$
14:    Update $f_{domain}$ with $\nabla \mathcal{L}_{domain}$
15:    **Invariant feature learning and alignment across domains**
16:    Compute: $\mathcal{L}_{cls}, \mathcal{L}_{domain}, \mathcal{L}_{global}, \mathcal{L}_{dtw}, \mathcal{L}_{center}$
17:    $\mathcal{L}_{total} = \mathcal{L}_{cls} - \lambda_{domain}\mathcal{L}_{domain} + \lambda_{global}\mathcal{L}_{global} + \lambda_{dtw}\mathcal{L}_{dtw} + \lambda_{center}\mathcal{L}_{center}$
18:    Update $f_g, f_l, g, f_{cls}$ with $\nabla \mathcal{L}_{total}$
19: **end for**

---

# E   Symbols and Notations

| Symbols | Notations |
|---------|-----------|
| $x^s$ | input from source domain |
| $x^t$ | input from target domain |
| $y^s$ | label from source domain |
| $y^t$ | label from target domain |
| $\mathcal{D}_S$ | source domain |
| $\mathcal{D}_T$ | target domain |
| $\mathcal{S}$ | source domain dataset |
| $\mathcal{T}$ | target domain dataset |
| $o$ | patch |
| $z_g$ | global representation |
| $z_l$ | local representation |
| $\hat{z}$ | fused representation |
| $c_j$ | the $j$th class prototype |
| $F(\cdot)$ | feature extractor |
| $G(\cdot)$ | fusion module |
| $C(\cdot)$ | classifier |
| $D(\cdot)$ | domain discriminator |
| $T$ | number of time steps |
| $N$ | number of different scale local encoders |
| $M$ | number of patches |
| $P$ | length of patch |
| $S$ | stride between adjacent patches |

Table 6: **Symbols and Notations**

# F ADDITIONAL EXPERIMENTS

## F.1 INFERENCE TIME AND PARAMETERS

We computed and compared the computational complexity and parameter count of LogoRA with other time series DA methods, as well as the general DA method CDAN. As illustrated in the Figure 8, our LogoRA incurs only a slight increase in computational complexity and the size of model parameters, but leads to a significant improvement in generalization (Sec. 4).

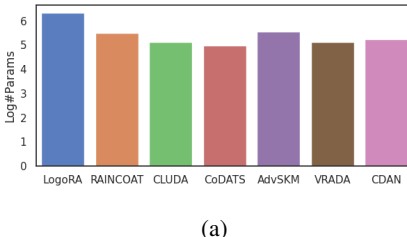

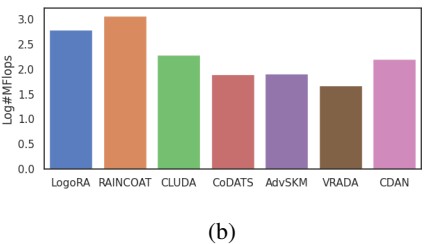

(a)                 (b)

Figure 8: **(a):** Logarithmically transformed parameter count of different methods **(b):** Logarithmically transformed million floating point operations per second of different methods

## F.2 EFFECT OF HYPER-PARAMETERS

We also conduct a series of experiments to study the impact of different hyperparameters on the results in Table 7. All experiments are conducted on the HHAR dataset. Here, we primarily investigate the impact of the following parameters: (1) $lr$: learning rate $lr$; (2) # Layers: the number of layers in the global encoder of LogoRA; (3) Patch length: the length of patch in the global encoder, where the corresponding stride is set half of it; (4) Kernels: the kernel sizes of different convolutional neural network in the multi-scale encoders. Here we all employ three networks with distinct kernel sizes. (5) $\lambda_{domain}$, the hyper-parameter of $\mathcal{L}_{domain}$; (6) $\lambda_{center}$, the hyper-parameter of $\mathcal{L}_{center}$; (7) $\lambda_{global}$, the hyper-parameter of $\mathcal{L}_{global}$; (8) $\lambda_{dtw}$, the hyper-parameter of $\mathcal{L}_{dtw}$. When changing the value of a specific parameter, the values of other parameters remain at their default values, as indicated in bold in the table.

Table 7: **Results of different LogoRA 's hyper-parameter configurations** on the HHAR dataset. $lr$ is the learning rate for the whole framework. # Layers is the number of layers in the global encoder of LogoRA. Kernels are the kernel sizes of the multi-scale local encoders, respectively. Patch length is the length of patch in the global encoder. The default values for all parameters are indicated in bold.

| Hyper-Parameter | Value | Accuracy | | | | Hyper-Parameter | Value | Accuracy | | | |
| --- | --- | --- | --- | --- | --- | --- | --- | --- | --- | --- | --- |
| | | $2 \mapsto 4$ | $7 \mapsto 1$ | $8 \mapsto 3$ | Avg | | | $2 \mapsto 4$ | $7 \mapsto 1$ | $8 \mapsto 3$ | Avg |
| $lr$ | 1.00E-02 | 0.195 | 0.149 | 0.170 | 0.201 | # Layers | 4 | 0.494 | 0.873 | 0.974 | 0.771 |
| | **1.00E-03** | 0.865 | 0.910 | 0.974 | 0.829 | | 6 | 0.478 | 0.907 | 0.965 | 0.767 |
| | 1.00E-04 | 0.657 | 0.619 | 0.764 | 0.635 | | **8** | 0.865 | 0.910 | 0.974 | 0.829 |
| Patch length | 8 | 0.940 | 0.840 | 0.974 | 0.803 | Kernels | **4-8-16** | 0.865 | 0.910 | 0.974 | 0.829 |
| | **16** | 0.865 | 0.910 | 0.974 | 0.829 | | 8-16-32 | 0.586 | 0.705 | 0.952 | 0.759 |
| | 32 | 0.590 | 0.858 | 0.830 | 0.794 | | 2-4-8 | 0.478 | 0.843 | 0.900 | 0.742 |
| $\lambda_{domain}$ | 1 | 0.462 | 0.914 | 0.821 | 0.712 | $\lambda_{center}$ | **0.1** | 0.865 | 0.910 | 0.974 | 0.829 |
| | **2** | 0.865 | 0.910 | 0.974 | 0.829 | | 0.2 | 0.558 | 0.881 | 0.969 | 0.753 |
| | 3 | 0.470 | 0.843 | 0.969 | 0.771 | | 0.3 | 0.219 | 0.336 | 0.969 | 0.606 |
| $\lambda_{global}$ | 0.3 | 0.478 | 0.806 | 0.974 | 0.774 | $\lambda_{dtw}$ | 0.3 | 0.936 | 0.914 | 0.943 | 0.847 |
| | **0.5** | 0.865 | 0.910 | 0.974 | 0.829 | | **0.5** | 0.865 | 0.910 | 0.974 | 0.829 |
| | 0.7 | 0.610 | 0.601 | 0.965 | 0.725 | | 0.7 | 0.470 | 0.765 | 0.200 | 0.760 |

As observed from Table 7, a learning rate of 1.00E-03 is the most suitable for our LogoRA. Therefore, all other experiments in this paper are based on this learning rate. We experiment with three different depths of transformers as the global encoder. The results indicate that the model structure with 8 layers is the most suitable. Next, we investigate the length of patch. When setting the patch length to 16, LogoRA gets the best result. Besides, we try three different types of multi-scale local encoder and find the encoder with kernel size of 4, 8, and 16 performs the best. After fixing the

parameters of model structure, We adjust the weights of different loss functions ($\lambda_{domain}$, $\lambda_{center}$, $\lambda_{global}$, and $\lambda_{dtw}$). It is obvious that different weights may suit for different scenarios (source domain $\mapsto$ target domain). Therefore, we try numerous possible weights of loss functions for all scenarios, and for each scenario, we report the best-performing result. Notably, all these results are based on HHAR dataset. If changing to other datasets, all the hyper-parameters need to be returned, which is detailed in Tab 5.

## F.3 T-SNE VISUALIZATIONS OF LEARNED REPRESENTATIONS

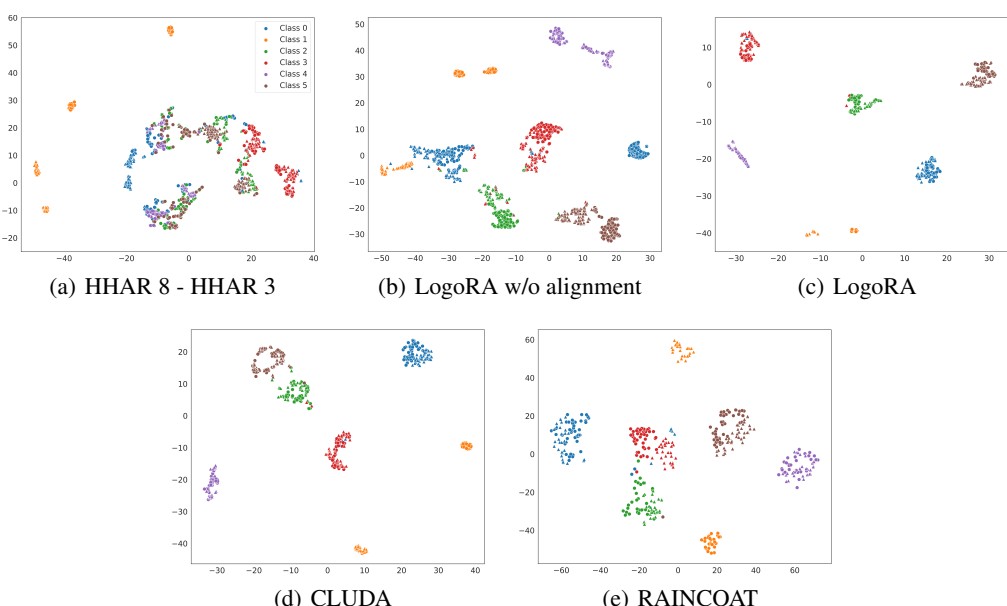

    (a) HHAR 8 - HHAR 3        (b) LogoRA w/o alignment        (c) LogoRA

             (d) CLUDA              (e) RAINCOAT

Figure 9: We generated T-SNE plots of learned embeddings for different methods and raw data on HHAR dataset of adapting from source 8 to target 3. In each plot, each color corresponds to a different class. The circle markers represent source samples, while the triangle represents target.

We generated T-SNE plots of learned embeddings for different methods. In Figure 9, we present the T-SNE plots of the original data, RAINCOAT (He et al., 2023), CLUDA (Ozyurt et al., 2023), LogoRA without any alignment and the complete LogoRA for HHAR dataset of adapting source 8 to target 3. Despite the original data being challenging to discern, LogoRA is still capable of effectively distinguishing between different categories and aligning the features of the target domain with the source domain. When no alignment is applied, *i.e.* the model is only trained on source domain, the domain gap between source domain and target domain is obvious. Compared to RAINCOAT and CLUDA, our clustering results are also more concentrated. Moreover, for Class 1, which experiences significant domain shift, LogoRA achieves a much better alignment than CLUDA and RAINCOAT. This suggests that LogoRA effectively adapts the model to the target domain, leading to improved performance and more accurate predictions. These findings also demonstrate the efficacy of LogoRA for domain adaptation and highlight its potential for a wide range of applications, including robotics, healthcare, and sports performance analysis.

## F.4 FAILURE CASE ANALYSIS

While the model exhibits strong transferability, there are still some misclassified samples, as shown in Figure 10. We present three-channel data for a pair of samples from the target domain and the source domain on the HAR dataset. It can be observed that when there is a substantial gap in features between the source and target domains, even LogoRA struggles to achieve successful alignment. The features in Channel 5 and Channel 6 are not as pronounced as in Channel 7. This could potentially lead to the failure of domain adaptation for multivariate time series data. Therefore, maybe stronger models are needed to learn more features from the multivariate data, as well as the development of more effective alignment methods.

## F.5 ADDITIONAL ABLATION STUDY

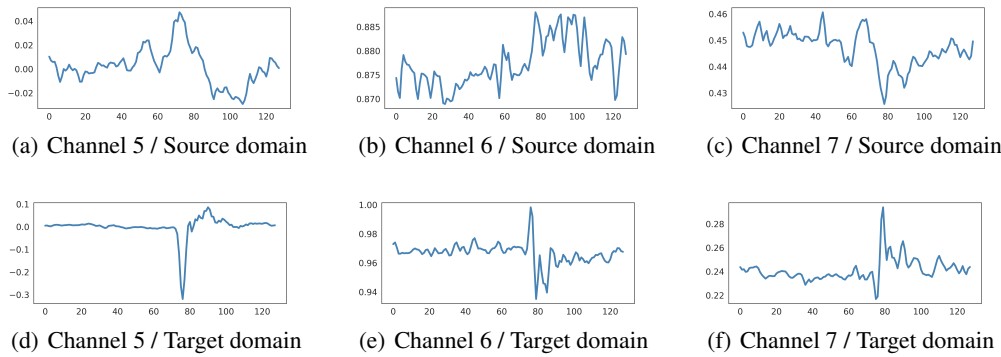

(a) Channel 5 / Source domain    (b) Channel 6 / Source domain    (c) Channel 7 / Source domain

(d) Channel 5 / Target domain    (e) Channel 6 / Target domain    (f) Channel 7 / Target domain

Figure 10: **Failure Case:** one of the misclassified samples, where we present three channels of data from the source domain and target domain.

Table 8: **Ablation studies of loss function on HAR and WISDM dataset.** Specifically, the loss functions, $\mathcal{L}_{cls}$, $\mathcal{L}_{domain}$, $\mathcal{L}_{global}$, $\mathcal{L}_{dtw}$, and $\mathcal{L}_{center}$, are shown below. When only the classification loss $\mathcal{L}_{cls}$ is used (first row), it refers to a source-only model, which is trained exclusively on the source domain. We evaluate LogoRA across 10 scenarios on the HAR and WISDM dataset and report the mean Accuracy.

| \multicolumn{5}{c}{Element of LogoRA} | | \multicolumn{2}{c}{UDA performance} | |
|---|---|---|---|---|---|---|
| $\mathcal{L}_{cls}$ | $\mathcal{L}_{domain}$ | $\mathcal{L}_{global}$ | $\mathcal{L}_{dtw}$ | $\mathcal{L}_{center}$ | WISDM | HAR |
| ✓ | | | | | 0.712 | 0.877 |
| ✓ | ✓ | | | | 0.786 | 0.923 |
| ✓ | | ✓ | ✓ | | 0.734 | 0.892 |
| ✓ | ✓ | | | ✓ | 0.791 | 0.928 |
| ✓ | ✓ | ✓ | | ✓ | 0.794 | 0.936 |
| ✓ | ✓ | | ✓ | ✓ | 0.813 | 0.911 |
| ✓ | ✓ | ✓ | ✓ | ✓ | **0.831** | **0.979** |

We additionally conduct ablation experiments of loss functions on the HAR and WISDM datasets, and report the average accuracy over 10 scenarios for each case. The experimental results are shown in the Table 8. Based on the results from Table 2 and Table 8, we can draw a consistent conclusion that each component in the entire framework is necessary and effective.

F.6 ADDITIONAL EXPERIMENTS ON OTHER DATASET

We expand our experiments to include an additional challenging dataset, CAP (Terzano et al., 2001), which has 3000 time steps and 19 dimensions. We tested our LogoRA and some other baselines on CAP, as shown in Table 9. Despite the complexity of the CAP dataset, where all methods show relatively low mean accuracy, our LogoRA still exhibits significant improvements over other baselines and achieves the best performance.

Table 9: **UDA performance on CAP dataset.** LogoRA consistently outperforms all other methods in accuracy on test sets drawn from the target domain dataset.

| Source $\mapsto$ Target | MMDA | CDAN | CLUDA | RAINCOAT | **LogoRA** |
|---|---|---|---|---|---|
| $0 \mapsto 1$ | 0.392 | 0.409 | 0.422 | 0.525 | **0.601** |
| $4 \mapsto 2$ | 0.271 | 0.277 | 0.271 | 0.317 | **0.359** |
| $4 \mapsto 3$ | 0.363 | 0.374 | 0.425 | 0.441 | **0.564** |
| $3 \mapsto 2$ | 0.230 | 0.343 | 0.264 | 0.343 | **0.357** |
| $0 \mapsto 4$ | 0.343 | 0.406 | 0.443 | 0.608 | **0.662** |
| $1 \mapsto 3$ | 0.249 | 0.368 | 0.449 | 0.475 | **0.489** |
| Average | 0.308 | 0.363 | 0.379 | 0.452 | **0.506** |

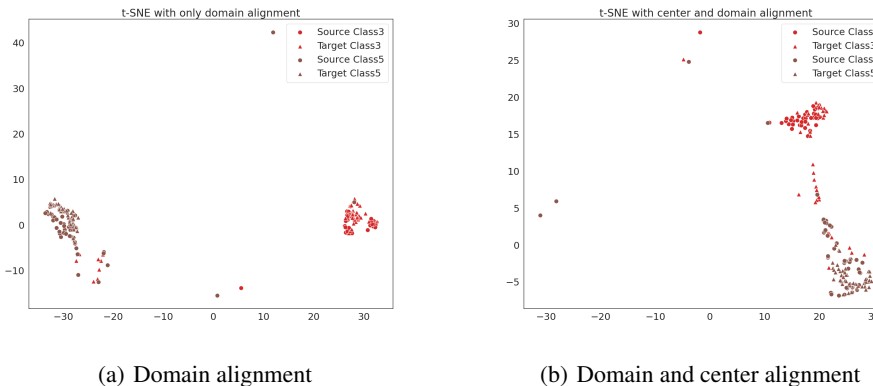

(a) Domain alignment         (b) Domain and center alignment

Figure 11: We generated T-SNE plots of learned embeddings for different alignments on HHAR dataset of adapting from source 2 to target 4.

### F.7 EXPLANATION ON THE PERFORMANCE DROP ON SPECIFIC DOMAIN PAIRS

As shown in the fourth row compared to the second row in Table 2, when the model further aligns target domain samples with source domain prototypes using center loss $\mathcal{L}_{center}$ after the initial domain alignment $\mathcal{L}_{domain}$, there is a slight improvement in most domain pairs. However, on $2 \mapsto 4$, the accuracy has a significant decrease. To further investigate the reasons behind this, we visualized the embeddings learned from $2 \mapsto 4$ data before and after adding center loss using t-SNE. From the visualization in Figure 11(a), it can be observed that before using center loss, some samples from target domain class 3 are closer to the prototype of source domain class 5, and positive samples within the same class are more distant. Therefore, after applying center loss, they are incorrectly aligned with class 5, as shown in Figure 11(b). This misalignment leads to a decrease in the final classification accuracy. In contrast to $2 \mapsto 4$, in other domain pairs, positive samples in the target domain are relatively more concentrated (e.g., as shown in Figure 9(b)). Therefore, adding center alignment on top of domain alignment can further improve performance. In the case of $2 \mapsto 4$, the proposed global alignment loss is needed to align positive samples from each class in the feature space. And it results in a significant improvement in LogoRA accuracy after adding the global alignment loss, as shown in the fifth row of Table 2.

## G   ADDITIONAL MODEL ANALYSIS

### G.1   CROSS ATTENTION BETWEEN GLOBAL AND LOCAL FEATURE

Firstly, the cross-attention operation is able to extract useful robust context information between global and local features and is commonly used in other fields (Zhu et al., 2022). Thanks to the modeling capability of Transformers for long dependencies, we extract the global features of the data and force the global features to be time-step invariant through DTW. The local features extracted by the convolutional network remain in temporal order, which can be matched with the patch-wise global feature. Moreover, a global feature summarizes the content of a sequence, often leading to a compact representation. Local features, on the other hand, comprise pattern information about specific regions, which are especially useful for classification. Generally speaking, global features are better at recall, while local features are better at precision (Cao et al., 2020). So, we assume that local features with more relevant information to global features are more important. As shown in the heatmap in Figure 6, the higher the similarity calculated between global and local features, the more useful the corresponding part is for classification. Therefore, based on similarity score, further selecting local features can yield features that contain more useful contextual information. Besides, with different kernel sizes, we obtain several local features of different scale. So, using similarity score to select the multi-scale local feature helps to extract the effective features on different positions more comprehensively.

## H  LIMITATIONS AND FUTURE WORK

Although our model has achieved good results on various datasets from different modalities, its performance on the recent time series OOD benchmark, WOODS (Gagnon-Audet et al., 2022), is not as satisfactory. This benchmark includes larger datasets like CAP (Terzano et al., 2001). After several attempts (Table 9), not only recent works like CLUDA and RAINCOAT, but also our framework faces challenges in handling such massive data. Effectively addressing high-dimensional data may be one of the future directions for research.

Moreover, time series data are often coupled with static data such as images or text. Therefore, LogoRA has the potential to be extended to a broader range of multi-modal data. Leveraging the powerful encoding capabilities of transformers, different variants of transformers (e.g., Vision Transformer (Dosovitskiy et al., 2020), BERT (Devlin et al., 2018)) can be employed to encode inputs from various modalities. Subsequently, fusion methods, such as the cross-attention used in this paper, can be designed for combining embeddings from different modalities. Furthermore, both global and local information is crucial in different modalities of data. Therefore, a structure similar to the 2-branch multi-scale architecture we proposed can also be applied to multi-modal data.

