# OpenReview forum: "LogoRA: Local-Global Representation Alignment for Robust Time Series Classification"
_ICLR.cc/2024/Conference — Submitted to ICLR 2024_

### Official Review · Reviewer_E2UE · 2023-10-20

**Soundness:** 3 good
**Presentation:** 3 good
**Contribution:** 3 good
**Rating:** 6
**Confidence:** 3

**Summary:**

The paper proposes the Local-Global Representation Alignment framework LogoRA for unsupervised domain adaptation of time series data. The paper uses a two-branch encoder to extract local and global features, and uses triplet loss for fine alignment and dynamic time warping-based feature alignment. The experimental results demonstrate the effectiveness of the proposed method.

**Strengths:**

The paper is well-written and easy to understand.

The paper is theoretically sound.

The experimental results show the usefulness of the method.

**Weaknesses:**

The author should explain TCN when it appears for the first time.

The author claims that cross-attention in local-global fusion model is used to integrate local and global representations, which should be explained in detail. The cross-attention operation computes the similarity between query and key, and selects values based on the similarity score. It is confusing to compute the similarity of global and local feature, and use this score to select the local feature. What is the explanation behind it?

The paper only uses DTW for global features, how about the local features?

**Questions:**

See weaknesses above

---

> ### Author Response · Authors · 2023-11-18
> **Response**
>
> We thank the reviewer for their time and feedback on our paper and provide our responses below.
>
> ---
>
> **Q1: Explain TCN**
>
> **A1:** Sorry for the confusion. We have updated the manuscript and provided detailed explanations of TCN along with references, hoping to address your concerns.
>
> ---
>
> **Q2: The cross-attention operation computes the similarity between query and key, and selects values based on the similarity score. It is confusing to compute the similarity of global and local feature and use this score to select the local feature. What is the explanation behind it?**
>
> **A2:** Firstly, the cross-attention operation is able to extract useful robust context information between global and local features. Thanks to the modeling capability of Transformers for long dependencies, we extract the global features of the data and force the global features to be time-step invariant through DTW. The local features extracted by the convolutional network remain in temporal order, which can be matched with the patch-wise global feature. Moreover, a global feature summarizes the content of a sequence, often leading to a compact representation. Local features, on the other hand, comprise pattern information about specific regions, which are especially useful for classification. Generally speaking, global features are better at recall, while local features are better at precision [1]. So, we assume that local features with more relevant information to global features are more important. As shown in the heatmap in Figure 6, the higher the similarity calculated between global and local features, the more useful the corresponding part is for classification. Therefore, based on similarity score, further selecting local features can yield features that contain more useful contextual information. Besides, with different kernel sizes, we obtain several local features of different scale. So, using similarity score to select the multi-scale local feature helps to extract the effective features on different positions more comprehensively.  Moreover, taking the HHAR dataset as an example, we have performed additional experiments by changing the fusion method from cross attention to addition and concatenation. The results are shown in the table below. From the experimental results, cross attention also proves to be the much more effective than others. We have updated the manuscript and provided more detailed explanations of the cross-attention operation in Appendix G.1. We hope it could answer your doubt.
>
> | source → target | addition | concatenation | cross attention |
> | --- | --- | --- | --- |
> | 0 → 2 | 0.6849 | 0.7857 | 0.8235 |
> | 1 → 6 | 0.8884 | 0.9163 | 0.9163 |
> | 2 → 4 | 0.4462 | 0.3386 | 0.9363 |
> | 4 → 0 | 0.262 | 0.2576 | 0.3886 |
> | 4 → 1 | 0.6679 | 0.5672 | 0.9627 |
> | 5 → 1 | 0.8582 | 0.8694 | 0.9851 |
> | 7 → 1 | 0.8918 | 0.8545 | 0.9478 |
> | 7 → 5 | 0.4402 | 0.5637 | 0.8147 |
> | 8 → 3 | 0.8996 | 0.9738 | 0.9738 |
> | 8 → 4 | 0.6175 | 0.7331 | 0.9681 |
> | Avg | 0.6656 | 0.6860 | 0.8717 |
>
> ---
>
> **Q3: The paper only uses DTW for global features, how about the local features?**
>
> **A3:**  Thanks to the patching operation, the obtained global feature is shorter than the local feature obtained by convolution. Because DTW involves dynamic programming algorithms, aligning the global feature with DTW can save more time. Moreover, we have also attempted to align the local feature with DTW, and the experimental results indicate that there is little difference between the two methods. Therefore, we chose the method with higher time efficiency.
>
> ---
>
> We hope that our responses have solved your problems. Thank you for your thoughtful comments and suggestions again.
>
> References:
>
> [1] Cao B, Araujo A, Sim J. Unifying deep local and global features for image search[C]//Computer Vision–ECCV 2020: 16th European Conference, Glasgow, UK, August 23–28, 2020, Proceedings, Part XX 16. Springer International Publishing, 2020: 726-743.

---

> > ### Author Response · Authors · 2023-11-21
> > **Thank you for the review! Have we clearly addressed the concerns?**
> >
> > We greatly appreciate the time you took to review our paper. In the rebuttal, we have presented more detailed explanation and experiments for the proposed framework.
> >
> > Due to the short duration of the author-reviewer discussion phase, we would appreciate your feedback on whether your main concerns have been adequately addressed. We are ready and willing to provide further explanations and clarifications if necessary. Thank you very much!

---

> ### Author Response · Authors · 2023-11-23
> **Response to Reviewer E2UE (before the end of discussion)**
>
> Dear Reviewer E2UE,
>
> Since the End of author/reviewer rebuttal is approaching, may we know if our response addresses your main concerns? If so, we kindly ask for your reconsideration of the score.
>
> Should you have any further advice on the paper and/or our rebuttal, please let us know and we will be more than happy to engage in more discussion and paper improvements.
>
> Thank you so much for devoting time to improving our work!

---

### Official Review · Reviewer_RUDS · 2023-10-30

**Soundness:** 2 fair
**Presentation:** 3 good
**Contribution:** 2 fair
**Rating:** 5
**Confidence:** 4

**Summary:**

This paper primarily introduces an effective model for unsupervised domain adaptation in time series classification tasks, which integrates both global and local features efficiently. This integration has led to a commendable performance in the realm of unsupervised domain adaptation.

**Strengths:**

1. Employing an attention mechanism to fuse global and local features is particularly intriguing. This approach offers a novel perspective on feature integration in the context of time series data.

2. The experimental results demonstrate the method's impressive performance.

**Weaknesses:**

1. The reason for using DTW to align the patch representations is not sufficient because the time-step shift occurs on the original sequence. Why not align the original sequence?

2. More limitations of the proposed method need to be discussed. For instance, in Table 2 of the ablation experiments, there is a noticeable performance degradation, particularly in the 4th and 5th rows compared to the 3rd row. This can be attributed to differences in performance across the first three scenarios. Can specific examples be used to illustrate this?

3. The training objective specified in Equation (5) involves five distinct loss functions, rendering it challenging to optimize the overall loss coherently during training. In essence, the presence of multiple training losses poses a challenge in comprehending the fundamental contributions of this paper, given the complexity associated with optimizing such a diverse set of loss functions.

4. The explanation for why the fusion of global and local features is effective is not clear. The interpretation of the heatmaps in Figure 6 is unclear as well. (1) How can different positions in the heatmaps be correlated with sequences? (2) Which weight activations in the heatmaps are meaningful? (3) How do different scales of heatmaps demonstrate their effectiveness?

**Questions:**

1. Why is the method of fusing global and local features of time series using an attention mechanism effective?

2. What are the merits and drawbacks of the approach introduced in this paper, in comparison to existing methods, concerning both runtime efficiency and the convergence behaviour of the model?

3. Please answer other questions in the Weaknesses.

---

> ### Author Response · Authors · 2023-11-18
> **Response [1/2]**
>
> We thank the reviewer for their time and feedback on our paper and provide responses below.
>
> ---
>
> **Q1: The reason for using DTW to align the patch representations is not sufficient because the time-step shift occurs on the original sequence. Why not align the original sequence?**
>
> **A1:** **Firstly,** we aim to learn time-step invariant features through the DTW loss, so alignment should be performed at the feature level rather than the raw data level. **Secondly,** due to the dynamic programming process in DTW, aligning with the entire sequence is computationally inefficient and slow. Moreover, it will lead to training failure due to the enormous time cost in practice. The table below shows the FLOPs of using patch representations and the entire representations, respectively. **Thirdly**, using the entire sequence needs the naive Transformer, which presented worse performance than PatchTST in Table 3. Therefore, we choose to using patch subsequences for DTW alignment finally. We hope this clarification could answer your doubt.
>
>
> | patch representations| entire representation|
> | --- | --- |
> | 622.94M | 54.74G |
>
>
> ---
>
> **Q2: More limitations of the proposed method need to be discussed.**
>
> **A2:** Thank you for your constructive suggestion. We have updated the manuscript and presented the detailed explanation with clear visualizations in Appendix F.7. We also shared some limitations in Appendix H. We hope it could address your concern.
>
> ---
>
> **Q3: The training objective specified in Equation (5) involves five distinct loss functions, rendering it challenging to optimize the overall loss coherently during training. In essence, the presence of multiple training losses poses a challenge in comprehending the fundamental contributions of this paper, given the complexity associated with optimizing such a diverse set of loss functions.**
>
> **A3:** **Firstly,** from the results of ablation experiments in Table 2, it can be observed that without these loss functions, the model's performance is mediocre. However, after optimizing with all loss functions, our method achieves a significant improvement. **Secondly,** the proposed loss functions complement each other. They can be directly optimized, and the entire optimization process is not complex, allowing for quick convergence. **In conclusion,** we believe this precisely demonstrates our contribution. The proposed loss functions can operate at different granularities, and ultimately contribute to the outstanding performance.
>
> ---
>
> **Q4: The explanation for why the fusion of global and local features is effective is not clear.**
>
> **A4:** Thank you for your feedback. We have updated the manuscript and provided detailed explanations of the cross-attention operation in Appendix G.1. We hope it could answer your doubt.
>
> ---
>
> **Q5: The interpretation of the heatmaps in Figure 6 is unclear as well. (1) How can different positions in the heatmaps be correlated with sequences? (2) Which weight activations in the heatmaps are meaningful? (3) How do different scales of heatmaps demonstrate their effectiveness?**
>
> **A5**: **(1)** Because our local encoder uses a shallow convolutional network with a limited receptive field, the obtained features maintain the same temporal order as the original sequence, preserving a certain correspondence. **(2)** The weights obtained through cross-attention represent the weights of global features on the selection of local features. The higher the similarity calculated between global and local features, the more useful the corresponding part is for classification. Therefore, based on this score, further selecting local features can yield features that contain more useful contextual information. **(3)** With different kernel sizes, we obtain several local features of different scales. As shown in Figure 6, the weights assigned to different scales of local features are not identical. So, using similarity score to select the multi-scale local feature helps to extract the effective features on different positions more comprehensively.

---

> ### Author Response · Authors · 2023-11-18
> **Response [2/2]**
>
> **Q6: What are the merits and drawbacks of the approach introduced in this paper, in comparison to existing methods, concerning both runtime efficiency and the convergence behaviour of the model?**
>
> **A6:** We present the comparison of computational complexity with other methods in Appendix F.1. Besides, we have updated the manuscript and analyzed limitations in Appendix H. Below is a portion of the training logs, hoping to alleviate your concerns about convergence.
>
> ```markdown
> Epoch: [12][ 4/10]	Time  1.135	Data  0.007	Loss Cls 3.0972e+00	Macro F1   0.23	Loss Domain 7.9497e-01 	Loss dtw 2.7484e-01	Loss global 3.7793e-01	Loss center 3.4145e-01	Loss TOTAL 5.0477e+00
> Epoch: [24][ 8/10]	Time  1.244	Data  0.008	Loss Cls 2.0759e+00	Macro F1   0.91	Loss Domain 7.2277e-01 	Loss dtw 2.9608e-01	Loss global 1.6139e-01	Loss center 1.0612e+00	Loss TOTAL 3.7608e+00
> Epoch: [37][ 4/10]	Time  1.335	Data  0.019	Loss Cls 1.8064e+00	Macro F1   1.00	Loss Domain 7.4017e-01 	Loss dtw 2.8416e-01	Loss global 9.3070e-03	Loss center 1.1092e+00	Loss TOTAL 3.5444e+00
> Epoch: [49][ 8/10]	Time  1.342	Data  0.005	Loss Cls 2.0430e+00	Macro F1   0.99	Loss Domain 7.2754e-01 	Loss dtw 2.1802e-01	Loss global 9.5955e-02	Loss center 8.5806e-01	Loss TOTAL 3.7131e+00
> Epoch: [62][ 4/10]	Time  1.017	Data  0.005	Loss Cls 1.7018e+00	Macro F1   1.00	Loss Domain 7.2700e-01 	Loss dtw 2.5508e-01	Loss global 1.3439e-03	Loss center 1.3648e+00	Loss TOTAL 3.4205e+00
> Epoch: [74][ 8/10]	Time  1.188	Data  0.045	Loss Cls 1.5397e+00	Macro F1   1.00	Loss Domain 6.9720e-01 	Loss dtw 1.9302e-01	Loss global 5.3767e-03	Loss center 7.5330e-01	Loss TOTAL 3.1138e+00
> Epoch: [87][ 4/10]	Time  1.120	Data  0.009	Loss Cls 1.6324e+00	Macro F1   0.99	Loss Domain 7.2536e-01 	Loss dtw 1.5999e-01	Loss global 3.7332e-02	Loss center 5.3994e-01	Loss TOTAL 3.2358e+00
> Epoch: [99][ 8/10]	Time  1.257	Data  0.007	Loss Cls 1.5311e+00	Macro F1   1.00	Loss Domain 6.0868e-01 	Loss dtw 1.7000e-01	Loss global 5.6056e-03	Loss center 5.3012e-01	Loss TOTAL 2.8893e+00
> Epoch: [112][ 4/10]	Time  1.388	Data  0.007	Loss Cls 1.5883e+00	Macro F1   0.99	Loss Domain 6.7380e-01 	Loss dtw 1.2501e-01	Loss global 2.1499e-02	Loss center 5.4366e-01	Loss TOTAL 3.0635e+00
> Epoch: [124][ 8/10]	Time  1.276	Data  0.007	Loss Cls 1.7020e+00	Macro F1   0.99	Loss Domain 5.4085e-01 	Loss dtw 1.8139e-01	Loss global 6.4419e-02	Loss center 4.2474e-01	Loss TOTAL 2.9491e+00
> ```
>
> ---
>
> We hope that our responses have solved your problems. Thank you for your thoughtful comments and suggestions again.

---

> ### Author Response · Authors · 2023-11-20
> **Thank you for the review! Have we clearly addressed the concerns?**
>
> We greatly appreciate the time you took to review our paper. In the rebuttal, we have clarified the contribution of our work, including the novel alignment strategy, and presented more detailed explanation and experiments for the proposed framework.
>
> Due to the short duration of the author-reviewer discussion phase, we would appreciate your feedback on whether your main concerns have been adequately addressed. We are ready and willing to provide further explanations and clarifications if necessary. Thank you very much!

---

> ### Author Response · Authors · 2023-11-23
> **Response to Reviewer RUDS (before the end of discussion)**
>
> Dear Reviewer RUDS,
>
> Since the End of author/reviewer rebuttal is approaching, may we know if our response addresses your main concerns? If so, we kindly ask for your reconsideration of the score.
>
> Should you have any further advice on the paper and/or our rebuttal, please let us know and we will be more than happy to engage in more discussion and paper improvements.
>
> Thank you so much for devoting time to improving our work!

---

### Official Review · Reviewer_MkgY · 2023-10-31

**Soundness:** 3 good
**Presentation:** 2 fair
**Contribution:** 2 fair
**Rating:** 5
**Confidence:** 4

**Summary:**

This paper proposes the LogoRA which does time series classification according to the fusion feature, in which the fusion features are obtained by the fusion module that integrates the local and global representations extracted by CNN encoder and transformer encoder, respectively. LogoRA then uses different representations for invariant feature learning, and adversarial approaches to narrow the gap between domains.

**Strengths:**

The structure of the article is well organized and the logic is clear.
In UDA, local and global information of time series are used in combination.

**Weaknesses:**

In terms of writing, the first appearance of TCN in the sentence " Most existing approaches employ a TCN as the backbone…" on the first page should not be an abbreviation, otherwise we do not understand what it is. It should be described like RNNs, LSTM, and CNNs in the previous paragraph, with complete expressions followed by abbreviations.
The spelling of "evaluation" in the last sentence of the second page is incorrect.
Table 2 L_ cdan doesn't know what it refers to.
There are other writing errors like this.
The model structure is commonly used, and the idea of using adversarial training is also common. The innovation of aligning local features with global features is not enough to support the entire paper.

**Questions:**

How to determine the patch length P? Is it a hyperparameter of the associated dataset, trainable, self defined, or universal?
In feature invariant learning, is there an independent category for patch when selecting input p of the same class and input n of a different class, or is it equivalent to the category of the entire sequence? If so, what is the difference between using a patch subsequence and the entire sequence for DTW? Similarly, when calculating the Euclidean distance of a sequence, why choose to calculate on the fused features? Can DTW be calculated on fused features?
How to explain the significant decrease in the effectiveness of the 2->4 experiment compared to the second and fourth rows of the ablation experiment?
After adding global loss in the fifth row of the ablation experiment, the effect of the 7->1 experiment decreased by nearly half compared to the fourth row. How to explain this?

---

> ### Author Response · Authors · 2023-11-18
> **Response [1/2]**
>
> We thank the reviewer for their time and feedback on our paper and provide responses below.
>
> ---
>
> **Q1: About the writing errors**
>
> **A1:** Apologize for any confusion caused and we have updated the manuscript to correct the relevant writing errors. Thank you very much for your feedback.
>
> ---
>
> **Q2: The model structure is commonly used, and the idea of using adversarial training is also common. The innovation of aligning local features with global features is not enough to support the entire paper.**
>
> **A2:** **Firstly,** although similar model structures are common in computer vision (CV) and natural language processing (NLP) domains, as far as we know, we are the first to propose such a 2-branch structure in the field of time series modeling. **Secondly,** to our knowledge, there is limited prior work in time series that simultaneously focuses on both global and local features of sequences. Besides, our proposed method, which integrates global features with local features at different scales, is a novel contribution in this domain. **Thirdly,** except for adversarial training, we have devised different novel alignment strategies to regularize two sets of distinct features, enabling the learning of more robust features for the UDA task. The exploration on these strategies, which constrain feature learning at different granularities, is also one of our contributions. **Finally**, our experimental results demonstrate the effectiveness of the entire approach, showing significant improvements over previous works across various datasets from different modalities.
>
> ---
>
> **Q3: How to determine the patch length P?**
>
> **A3:** Following PatchTST [1], the patch length P is a self-defined hyperparameter obtained through grid search.
>
> ---
>
> **Q4: In feature invariant learning, is there an independent category for patch when selecting input p of the same class and input n of a different class, or is it equivalent to the category of the entire sequence?**
>
> **A4:** Sorry, we don't quite understand the specific issue you're referring to. We guess you might have some doubts about the triplet loss. Triplet loss is a metric learning loss where a reference input (called anchor) is compared to a matching input (called positive, i.e. p) and a non-matching input (called negative, i.e. n). The distance from the anchor to the positive is minimized, and the distance from the anchor to the negative input is maximized. The entire selection process is done within the same batch. Positive and negative samples are selected based on the labels of each sample. More detailed explanations can be found in reference [2]. Hope this clarification could address your confusion. Also, if you have any other questions, we are willing to answer your questions at any time.
>
> ---
>
> ---
>
> **Q5: What is the difference between using a patch subsequence and the entire sequence for DTW?**
>
> **A5**: **Firstly**, the essential difference between the two methods is not significant. However, due to the dynamic programming process in DTW, aligning with the entire sequence is computationally inefficient. Moreover, it will lead to training failure due to the enormous time cost in practice. The table below shows the FLOPs of using patch representations and the entire representations, respectively. **Secondly**, using the entire sequence needs the naive Transformer, which exhibits worse performance than PatchTST in Table 3. Therefore, we choose to using patch subsequences for DTW alignment finally. We hope this clarification could answer your doubt.
>
> | patch representations| entire representation|
> | --- | --- |
> | 622.94M | 54.74G |

---

> ### Author Response · Authors · 2023-11-18
> **Response [2/2]**
>
> **Q6: When calculating the Euclidean distance of a sequence, why choose to calculate on the fused features?**
>
> **A6: Firstly,** the final fused feature is input into the classifier for classification, so we aim to further align the features corresponding to samples of the same class in the source domain in the feature space. **Secondly,** as shown in the t-SNE visualization in Figure 9, the aligned features are more clustered. It will facilitate the further use of center loss to align the features of the target domain with the source domain prototypes that are closest to them in the feature space.
>
> ---
>
> **Q7:** **Can DTW be calculated on fused features?**
>
> **A7:** Due to the fact that in our proposed Local-Global Fusion Module, the final features are summed over the temporal dimension for classification. As a result, the fused feature no longer has temporal dimensions. So, it is impractical to compute DTW on the fused features.
>
> ---
>
> **Q8: How to explain the significant decrease in the effectiveness of the 2->4 experiment compared to the second and fourth rows of the ablation experiment?**
>
> **A8:** Thank you for your feedback. We have updated the manuscript and presented the detailed explanation with clear visualizations in Appendix F.7. We hope it could address your concern.
>
> ---
>
> **Q9: After adding global loss in the fifth row of the ablation experiment, the effect of the 7->1 experiment decreased by nearly half compared to the fourth row. How to explain this?**
>
> **A9:** We have modified the random seed and reimplemented the experiments three times for scenario 7-1. The results, as shown in the table below, indicate that the experimental variance on this domain pair reaches 0.03. Further investigation reveals that the test set for domain 7 has only 1116 samples, and for domain 1, there are only 268 samples. Therefore, we believe that the randomness in the experimental results is largely due to the insufficient amount of data.  We have updated the manuscript and presented the average experimental results.
>
> | Run1 | Run2 | Run3 | Original result | Average |
> | --- | --- | --- | --- | --- |
> | 0.7537 | 0.8843 | 0.6754 | 0.4701 | 0.6959 |
>
> ---
>
> We hope that our responses have solved your problems. Thank you for your thoughtful comments and suggestions again.
>
> References:
>
> [1] Nie Y, Nguyen N H, Sinthong P, et al. A Time Series is Worth 64 Words: Long-term Forecasting with Transformers[C]//The Eleventh International Conference on Learning Representations. 2022.
>
> [2] Schroff F, Kalenichenko D, Philbin J. Facenet: A unified embedding for face recognition and clustering[C]//Proceedings of the IEEE conference on computer vision and pattern recognition. 2015: 815-823.

---

> ### Author Response · Authors · 2023-11-20
> **Thank you for the review! Have we clearly addressed the concerns?**
>
> We greatly appreciate the time you took to review our paper. In the rebuttal, we have clarified the contribution of our work, including the new model structure and novel alignment strategy, and presented more detailed explanation and experiments for the proposed framework.
>
> Due to the short duration of the author-reviewer discussion phase, we would appreciate your feedback on whether your main concerns have been adequately addressed. We are ready and willing to provide further explanations and clarifications if necessary. Thank you very much!

---

> ### Author Response · Authors · 2023-11-23
> **Response to Reviewer MkgY (before the end of discussion)**
>
> Dear Reviewer MkgY,
>
> Since the End of author/reviewer rebuttal is approaching, may we know if our response addresses your main concerns? If so, we kindly ask for your reconsideration of the score.
>
> Should you have any further advice on the paper and/or our rebuttal, please let us know and we will be more than happy to engage in more discussion and paper improvements.
>
> Thank you so much for devoting time to improving our work!

---

### Official Review · Reviewer_oA7T · 2023-11-02

**Soundness:** 4 excellent
**Presentation:** 3 good
**Contribution:** 3 good
**Rating:** 6
**Confidence:** 4

**Summary:**

This paper presents a universal framework for unsupervised (actually semi-supervised) sequence modeling that captures both local patterns and global patterns. Instead of using simple concatenation, it proposes to used cross attention, which has been shown promising in recent studies. Many regularization losses have been included to enforce alignment.

**Strengths:**

Overall, this paper is well-polished and experiments are extensive. More specifically,
1: The ablation study is very comprehensive. The table 2 and 3 provides very convincing evidence for each component and why they should be part of the framework.
2: The choice of loss functions is convincing. I can see that authors have put many thoughts on it. Table 2 is a strong support for each loss function.
3: Authors have included many baselines in the study, including models from recent papers.

**Weaknesses:**

1: I am interested in the implementation, as reproducibility is a key metric in ML publications nowadays. I wish authors can provide a code repo (hopefully using notebook so I can see the results without re-running everything). If authors can show the reproducibility durign rebuttal, I would like to increase my rating.
2: The cross-attention is only validated by qualitative visualizations but not in ablation study. Although the heatmap looks good to me, it is interesting to know how much gain does cross attention bring vs. other fusion methods (e.g., addition or concatenation).
3: As the authors have ackowledged in the last paragraph, this framework does not work well in other public datasets, so this paper most likely only includes those successful datasets. I would invite the authors to append their explorations on other datasets in the appendix or code repo, so others (like me) can learn from it and investigate further.

**Questions:**

Usually time series data are coupled with static data (tabular, images or texts), for example, MIMIC dataset. How to expand the current framework to deal with multi-modal data? You can share your thoughts as limitation/future work. If you have tried, you could share your exploration in the appendix too (even if it failed).

---

> ### Author Response · Authors · 2023-11-18
> **Response**
>
> We thank the reviewer for the constructive feedback and provide responses below.
>
> ---
>
> **Q1: About the reproducibility**
>
> **A1:** The code repository is now publicly accessible via the following [anonymous link](https://anonymous.4open.science/r/LogoRA_rebuttal-DBE4/). Additionally, we have prepared a Jupyter notebook to facilitate the verification of the reproducibility of our experiments. The notebook is included in the code repository and can be accessed directly through the provided link. This is not the final official repo because we want to maintain anonymity. Once the anonymous review ends we will update it with a final official link.
>
> ---
>
> **Q2: Cross attention vs. other fusion methods**
>
> **A2:** We have performed additional experiments by changing the fusion method from cross attention to addition and concatenation. The results are shown in the table below. Using cross attention is much more effective in fusing global features and local features than directly adding or concatenating them. As shown in the heatmap in Figure 6, by using the weights calculated between the global features, which have already learned time-step invariant information, and the local features, it is possible to focus on the parts of local features that are more useful for classification and learn more robust context information.
>
> | source → target | addition | concatenation | cross attention |
> | --- | --- | --- | --- |
> | 0 → 2 | 0.6849 | 0.7857 | 0.8235 |
> | 1 → 6 | 0.8884 | 0.9163 | 0.9163 |
> | 2 → 4 | 0.4462 | 0.3386 | 0.9363 |
> | 4 → 0 | 0.262 | 0.2576 | 0.3886 |
> | 4 → 1 | 0.6679 | 0.5672 | 0.9627 |
> | 5 → 1 | 0.8582 | 0.8694 | 0.9851 |
> | 7 → 1 | 0.8918 | 0.8545 | 0.9478 |
> | 7 → 5 | 0.4402 | 0.5637 | 0.8147 |
> | 8 → 3 | 0.8996 | 0.9738 | 0.9738 |
> | 8 → 4 | 0.6175 | 0.7331 | 0.9681 |
> | Avg | 0.6656 | 0.6860 | 0.8717 |
>
> ---
>
> **Q3: Future work and limitations**
>
> **A3:** We thank the reviewer for useful suggestions. We presented several analysis on failure cases in Appendix F.4. And we have updated the manuscript and shared some new attempts in Appendix F.6, along with some ideas on how to extend our framework to multi-modal data in Appendix H. We hope these insights could be valuable to you. As for the MIMIC dataset you mentioned, unfortunately, our application for access is not replied by the official authorities. Consequently, we cannot to conduct experiments on this dataset. However, we express our willingness to explore its effects in the future if our application is approved. You are also welcome to try our publicly available code on your own. We hope it is helpful for you.
>
> ---
>
> We hope that our responses have solved your problems. Thank you for your thoughtful comments and suggestions again.

---

> ### Author Response · Authors · 2023-11-20
> **Thank you for the review! Have we clearly addressed the concerns?**
>
> We greatly appreciate the time you took to review our paper. In the rebuttal, we have provided the code repository as evidence of our reproducibility and presented more detailed future work and experiments for the proposed framework.
>
> Due to the short duration of the author-reviewer discussion phase, we would appreciate your feedback on whether your main concerns have been adequately addressed. We are ready and willing to provide further explanations and clarifications if necessary. Thank you very much!

---

> ### Author Response · Authors · 2023-11-23
> **Response to Reviewer oA7T (before the end of discussion)**
>
> Dear Reviewer oA7T,
>
> Since the End of author/reviewer rebuttal is approaching, may we know if our response addresses your main concerns? If so, we kindly ask for your reconsideration of the score.
>
> Should you have any further advice on the paper and/or our rebuttal, please let us know and we will be more than happy to engage in more discussion and paper improvements.
>
>
> Thank you so much for devoting time to improving our work!

---

### Official Review · Reviewer_UBmN · 2023-11-04

**Soundness:** 2 fair
**Presentation:** 3 good
**Contribution:** 2 fair
**Rating:** 3
**Confidence:** 5

**Summary:**

This paper studies the problem of unsupervised domain adaptation on time series data in classification, which is to help ML models to adapt the other scenarios. The authors proposes LogoRA to extract both local and global representations. The result indicates a improvement of 12% on four datasets.

**Strengths:**

1.The authors design a new metric learning method based on DTW, which can overcome the severe time-shift patterns that exist in time series data and learn more robust features from the source domain.

2.The proposed method extract both global and local features different domains and  outperforms baselines by up to 12.52%

**Weaknesses:**

1. The technical details seem to be a bit lack of innovation. The author can improve the paper by further explain how global and local information work together to improve the quality of the classification.

2. Only four datasets are employed, it may be not sufficient enough to support the claim of the advantage of the proposed method.

3. Since the ablation experiment was only conducted on the best performing HHAR dataset, we believe that the results of the ablation experiment do not fully support the author's work

**Questions:**

1. The effectiveness of the algorithm proposed by the author compared to the baseline method varies significantly across four datasets, with an improvement of+12.52% on the HHAR dataset, but only+0.51% on the HAR dataset. Can the author explain the reason for this difference?

2. The author suggests that using both global and local information simultaneously can help improve algorithm quality. Can we use intuitive examples to explain this process?

---

> ### Author Response · Authors · 2023-11-18
> **Response [1/2]**
>
> We thank the reviewer for the constructive feedback and provide responses below.
>
> ---
>
> **Q1: The author can improve the paper by further explain how global and local information work together to improve the quality of the classification.**
>
> **A1:** We thank the reviewer for useful suggestions. Firstly, the cross-attention operation is able to extract useful robust context information between global and local features. Thanks to the modeling capability of Transformers for long dependencies, we extract the global features of the data and force the global features to be time-step invariant through DTW. The local features extracted by the convolutional network remain in temporal order, which can be matched with the patch-wise global feature. As shown in the heatmap in Figure 6, the higher the similarity calculated between global and local features, the more useful the corresponding part is for classification. Therefore, based on similarity score, further selecting local features can yield features that contain more useful contextual information. Besides, with different kernel sizes, we obtain several local features of different scale. So, using similarity score to select the multi-scale local feature helps to extract the effective features on different positions more comprehensively. Besides, we have performed additional experiments by changing the fusion method from cross attention to addition and concatenation. The results are shown in the table below. It can be observed that using cross attention is much more effective than other fusion methods. We have also updated the manuscript and provided detailed explanations of the cross-attention operation in Appendix G.1.
>
> | source → target | addition | concatenation | cross attention |
> | --- | --- | --- | --- |
> | 0 → 2 | 0.6849 | 0.7857 | 0.8235 |
> | 1 → 6 | 0.8884 | 0.9163 | 0.9163 |
> | 2 → 4 | 0.4462 | 0.3386 | 0.9363 |
> | 4 → 0 | 0.262 | 0.2576 | 0.3886 |
> | 4 → 1 | 0.6679 | 0.5672 | 0.9627 |
> | 5 → 1 | 0.8582 | 0.8694 | 0.9851 |
> | 7 → 1 | 0.8918 | 0.8545 | 0.9478 |
> | 7 → 5 | 0.4402 | 0.5637 | 0.8147 |
> | 8 → 3 | 0.8996 | 0.9738 | 0.9738 |
> | 8 → 4 | 0.6175 | 0.7331 | 0.9681 |
> | Avg | 0.6656 | 0.6860 | 0.8717 |
>
> ---
>
> **Q2: Only four datasets are employed, it may be not sufficient enough to support the claim of the advantage of the proposed method.**
>
> **A2:** We acknowledge the importance of a diverse set of datasets for a comprehensive evaluation of our proposed method. Therefore, following recent work in time series domain adaptation (reference [1] and reference [2]), we employed a similar amount of datasets for a fair comparison. Certainly, we understand your concern, so we have expanded our experiments to include an additional dataset, CAP (detailed in reference [3]). Compared to the previous datasets, the time series in the CAP have longer lengths and more dimensions, making the dataset more challenging. Due to time constraints, we only test a portion of the baselines, which are reported in the Appendix F.6. The results presented in Table 9 further demonstrate that our method has significant improvements over previous works, achieving better results on more complex datasets. We hope this could address your concerns.
>
> ---
>
> **Q3: The ablation experiment was only conducted on the HHAR dataset.**
>
> **A3:** Thank the reviewer for constructive feedback. We would like to note that our ablation study provides very convincing evidence for each component and why they should be part of the framework, which is appreciated by Reviewer oA7T and E2UE. Of course, we understand your concerns, so we have updated the manuscript and presented the ablation study on other datasets in Appendix F.5. We believe that the whole ablation experiments are able to prove the effectiveness of our work and hope it could address your concerns.

---

> ### Author Response · Authors · 2023-11-18
> **Response [2/2]**
>
> **Q4: The effectiveness of the algorithm proposed by the author compared to the baseline method varies significantly across four datasets, with an improvement of+12.52% on the HHAR dataset, but only+0.51% on the HAR dataset. Can the author explain the reason for this difference?**
>
> **A4:** **Firstly**, it should be noted that the performance on the HAR dataset has been saturated. For example, the current state-of-the-art RAINCOAT achieves 0.974 in terms of mean accuracy. So, there is a little room for improvement. Our work pushes that boundary without 2-stage training or any correction steps in the reference [2], achieving a 0.51% absolute improvement over RAINCOAT and 3.71% absolute improvement over CLUDA, which has been a relative significant improvement on the HAR dataset. **Secondly**, in most testing cases on the HAR dataset, our method achieves excellent results with mean accuracy of 1.000. Especially, we achieve a 3.62% improvement on the 23-13 domain pair. **Thirdly**, the domain gaps between different datasets often vary depending on various data collection environments and application backgrounds, and the time-step shifts are also different. Therefore, it is quite natural that the degree of improvements achieved by our method vary across different datasets.
>
> ---
>
> **Q5: The author suggests that using both global and local information simultaneously can help improve algorithm quality. Can we use intuitive examples to explain this process?**
>
> **A5:** Simultaneously utilizing global and local information is a common and effective approach in various domains, such as computer vision (CV) as referenced in reference [4] and the time series domain as mentioned in reference [5]. Actually, we have presented an intuitive example as shown in Figure 1 in the section of Introduction. The example exhibits accelerometer data pieces of two actions, i.e., walking upstairs (upper) and walking downstairs (lower) from the HAR dataset. As shown in the lower figure in Figure1, compared to other regions, the local abrupt acceleration changes in the shaded area are more representative to characterize the class (walking downstairs), which underscores the significance of local features. Because the signal is relatively stable in other positions, it may be unable to capture this sudden signal change only using the global features obtained from the original data. This oversight could lead to a failure in recognizing this action. By contrast, if only using the local features in the shaded area in Figure 1, it is still not sufficient to distinguish the two sequences of walking downstairs and walking upstairs, as both sequences have similar local features. While it will be much easier to differentiate them through modeling the temporal dependencies between local features. Therefore, it is an optimal strategy to utilize global and local features simultaneously for improving the discriminative capability of time series models.
>
> ---
>
> We hope that our responses have solved your problems. Thank you for your thoughtful comments and suggestions again.
>
> References:
>
> [1] Ozyurt Y, Feuerriegel S, Zhang C. Contrastive Learning for Unsupervised Domain Adaptation of Time Series[C]//The Eleventh International Conference on Learning Representations. 2022.
>
> [2] He H, Queen O, Koker T, et al. Domain Adaptation for Time Series Under Feature and Label Shifts[J]. arXiv preprint arXiv:2302.03133, 2023.
>
> [3] Mario Giovanni Terzano, Liborio Parrino, Adriano Sherieri, Ronald Chervin, Sudhansu Chokroverty, Christian Guilleminault, Max Hirshkowitz, Mark Mahowald, Harvey Moldofsky, Agostino Rosa, et al. Atlas, rules, and recording techniques for the scoring of cyclic alternating pattern (cap) in human sleep. Sleep medicine, 2(6): 537–553, 2001.
>
> [4] Zhu H, Ke W, Li D, et al. Dual cross-attention learning for fine-grained visual categorization and object re-identification[C]//Proceedings of the IEEE/CVF Conference on Computer Vision and Pattern Recognition. 2022: 4692-4702.
>
> [5] Wang H, Peng J, Huang F, et al. Micn: Multi-scale local and global context modeling for long-term series forecasting[C]//The Eleventh International Conference on Learning Representations. 2022.

---

> ### Author Response · Authors · 2023-11-20
> **Thank you for the review! Have we clearly addressed the concerns?**
>
> We greatly appreciate the time you took to review our paper.  In the rebuttal, we have clarified the contribution of our work and provided more detailed explanation and experiments for the proposed framework.
>
> Due to the short duration of the author-reviewer discussion phase, we would appreciate your feedback on whether your main concerns have been adequately addressed. We are ready and willing to provide further explanations and clarifications if necessary. Thank you very much!

---

> ### Author Response · Authors · 2023-11-21
> **Response to Reviewer UBmN (before the end of discussion)**
>
> Dear Reviewer UBmN,
>
> Since the End of author/reviewer discussions is just in one day, may we know if our response addresses your main concerns? If so, we kindly ask for your reconsideration of the score.
>
> Should you have any further advice on the paper and/or our rebuttal, please let us know and we will be more than happy to engage in more discussion and paper improvements. We would really appreciate it if our next round of communication could leave time for us to resolve any of your remaining or new questions.
>
> Thank you so much for devoting time to improving our methodology!

---

> ### Author Response · Authors · 2023-11-23
> **Response to Reviewer UBmN (before the end of discussion)**
>
> Dear Reviewer UBmN,
>
> Since the End of author/reviewer rebuttal is approaching, may we know if our response addresses your main concerns? If so, we kindly ask for your reconsideration of the score.
>
> Should you have any further advice on the paper and/or our rebuttal, please let us know and we will be more than happy to engage in more discussion and paper improvements.
>
> Thank you so much for devoting time to improving our work!

---

### Comment · Area_Chair_bJ91 · 2023-11-22
**Pls provide your response to authors' feedback asap**

Dear All,

The authors have dedicated significant efforts to provide detailed responses. I would appreciate it if you could share your feedback with them.

Thank you for your valuable contributions to ICLR!

Best regards,
Area Chair

---

### Meta-Review · Area_Chair_bJ91 · 2023-12-07

**Metareview:**

(a) Scientific Claims and Findings:
The paper introduces LogoRA, a framework for unsupervised domain adaptation in time series classification. It emphasizes extracting both local and global representations, integrating features via DTW-based metric learning, and using cross-attention for fusion. Results show a 12% improvement across four datasets. However, disparities in performance among datasets raise concerns about method efficacy.

(b) Strengths:
1. Innovative Metric Learning: DTW-based metric learning addresses time-shift patterns in time series data, enhancing feature robustness.
2. Global and Local Feature Extraction: LogoRA's ability to extract and combine global and local features surpasses baselines by up to 12.52%.

(c) Weaknesses and Missing Aspects:
1. Lack of Innovation in Technical Details: Reviewers noted a need for better explanations regarding how global and local information collaboratively enhance classification quality.
2. Limited Experimental Scope: The use of only four datasets may not adequately support the claimed method advantage. A broader dataset analysis could strengthen claims.
3. Reproducibility and Validation: Lack of a code repository affects reproducibility. The validation of cross-attention through quantitative measures and exploration on diverse datasets are missing.

Overall, refining explanations, expanding the dataset analysis, providing a code repository for reproducibility, and addressing the reviewers' queries would significantly strengthen the paper.

**Justification For Why Not Higher Score:**

The reviewers appreciated several aspects of the paper, like the novel approach to feature extraction and the comprehensive experimentation. However, they highlighted a few key areas for improvement:

1. Clarity on Methodology:Reviewers asked for a clearer explanation of how global and local information integration works in improving classification quality. They also suggested providing more insights into the alignment of features.

2. Experimental Scope: Some reviewers pointed out the limitation in the number of datasets used for validation, suggesting that a broader set could better support the claimed advantages of the proposed method.

3. Reproducibility and Details: Addressing concerns about code availability, the validation of the proposed cross-attention methodology, and expanding the analysis to cover a broader spectrum of datasets were raised by reviewers as areas that could enhance the paper's credibility.

In essence, refining the explanations, extending the experimental scope, and providing additional details for validation and reproducibility were areas flagged for improvement.

**Justification For Why Not Lower Score:**

N/A

---

### Decision · Program_Chairs · 2024-01-16

Reject